# ENERGY-INSPIRED MOLECULAR CONFORMATION OPTIMIZATION

**Jiaqi Guan**[*]
University of Illinois at Urbana-Champaign
jiaqi@illinois.edu

**Wesley Wei Qian**[*]
University of Illinois at Urbana-Champaign
weiqian3@illinois.edu

**Qiang Liu**
University of Texas at Austin
lqiang@cs.utexas.edu

**Wei-Ying Ma**
AIR, Tsinghua University
maweiying@air.tsinghua.edu.cn

**Jianzhu Ma**
Peking University
Beijing Institute for General Artificial Intelligence
majianzhu@pku.edu.cn

**Jian Peng**
University of Illinois at Urbana-Champaign
AIR, Tsinghua University
HeliXon Limited
jianpeng@illinois.edu

## ABSTRACT

This paper studies an important problem in computational chemistry: predicting a molecule's spatial atom arrangements, or a molecular conformation. We propose a neural energy minimization formulation that casts the prediction problem into an unrolled optimization process, where a neural network is parametrized to learn the gradient fields of an implicit conformational energy landscape. Assuming different forms of the underlying potential energy function, we can not only reinterpret and unify many of the existing models but also derive new variants of SE(3)-equivariant neural networks in a principled manner. In our experiments, these new variants show superior performance in molecular conformation optimization comparing to existing SE(3)-equivariant neural networks. Moreover, our energy-inspired formulation is also suitable for molecular conformation generation, where we can generate more diverse and accurate conformers comparing to existing baselines.

## 1 INTRODUCTION

The 3D molecular conformation is one of the most important features in determining many physico-chemical and biological properties of a molecule. The molecule's 3D charge distribution and physical shape are crucial when considering the steric constraints or electronic effects for chemical reactions and interaction. Therefore, molecular conformers are widely adopted in quantitative structure-activity relationships (QSAR) prediction (Verma et al., 2010) and drug discovery (Hawkins, 2017). With a growing interest in virtual drug screening (Zhavoronkov et al., 2019; McCloskey et al., 2020; Stokes et al., 2020) and *de novo* drug design (De Cao & Kipf, 2018; Gómez-Bombarelli et al., 2018; Jin et al., 2018; Zhou et al., 2019; Mercado et al., 2020), it becomes more desirable to predict the molecular conformations both quickly *and* accurately such that 3D geometry features can be considered.

Deep learning techniques have recently been introduced to address this problem and demonstrated promising results. For instance, generative models (Hoffmann & Noé, 2019; Simm & Hernandez-Lobato, 2020; Xu et al., 2020) are proposed to first generate pairwise distances for atom pairs, and then infer the conformer from such matrix.Since the 3D structures are generated indirectly, the second step can be sensitive towards the error in estimated distances, and extra model capacity could be required to encode the redundancy (Hoffmann & Noé, 2019). In the meantime, many SE(3)-equivariant networks emerge to work with 3D roto-translation symmetrical inputs enabling direct manipulation of the 3D Cartesian coordinates. One of the most recent successful applications

---

[*]Equal Contributions.

of such models is the AlphaFold system for protein structure prediction (Jumper et al., 2021). Despite the many benefits, however, existing SE(3)-equivariant networks are either derived from complicated mathematical theory involving expensive coefficient calculations (Thomas et al., 2018; Anderson et al., 2019; Finzi et al., 2020; Fuchs et al., 2020), or heuristically designed (Satorras et al., 2021) based on the intuition of message passing (Gilmer et al., 2017).

This work presents a new formulation that unrolls the molecular conformer prediction into a fixed number of optimization steps, where a neural network is parametrized to learn gradient fields of the implicit conformational energy landscape. Under such formulation, the model can refine a given conformer toward a more stable and energetically preferred state. Instead of relying on complex mathematical tools or designing the model heuristically, we propose a novel energy-inspired framework that explicitly connects an underlying enregy function with an SE(3)-equivariant model. We are able to not only provide new interpretations for some existing models but also derive new variants of SE(3)-equivariant model from the assumption of a conformational energy function, aligning a model's architecture with its physical assumptions. Through extensive experiments, these new variants show superior performance in conformer *optimization* comparing to other SE(3)-equivariant models. Moreover, the proposed optimization method is also suitable for conformer *generation* and capable of generating diverse and accurate conformers.

In conclusion, the main contributions of this work include:

- Formulating the conformer prediction problem as a multi-step unrolled optimization where a model learns the gradient field of the conformational energy landscape.
- Proposing a novel framework that explicitly connects an underlying energy function with an SE(3)-equivariant neural network such that one can derive new models that align with a physical assumption, or interpret some of the existing models from an energy perspective.
- Demonstrating competitive empirical performance for molecular conformation prediction in both the optimization and generation settings.

## 2 RELATED WORK

### 2.1 MOLECUALR CONFORMATION GENERATION

In conventional computational approaches for conformer generation, we observe an accuracy and efficiency trade-off where highly accurate *ab initio* methods are very time-consuming while efficient methods leveraging empirical force fields are not accurate enough (Unke et al., 2021). To efficiently *and* accurately estimates molecular conformations, many learning-based methods have been proposed. For instance, models, such as VAE (Simm & Hernandez-Lobato, 2020), GAN (Hoffmann & Noé, 2019), and continuous normalizing flow (Xu et al., 2020), have all been developed to predict molecular conformers by first generating an intermediate pairwise atom *distance matrix*. However, the post-processing step for 3D structure construction can be susceptible to the error, and model capacity could be wasted to encode the redundancy in such low-rank matrix (Hoffmann & Noé, 2019). Despite the many drawbacks of such methods, the distance matrix is difficult to circumvent because the models cannot capture the 3D roto-translation symmetry nature of a conformer.

Besides modeling the atom distances, another solution for conformer generation is to estimate the *force field* with machine learning methods. For instance, sGDML (Chmiela et al., 2019) is capable of reproducing the molecular potential energy surface given a few reference conformations and generating compelling results with MD simulation. Although such a model scales well with the molecule size, it requires retraining for every new molecules (Unke et al., 2021). Similarly but more generally, methods including Behler & Parrinello (2007); Bartók et al. (2010; 2013); Smith et al. (2017); Schütt et al. (2017); Faber et al. (2018) can also be used to generate molecular conformers with their learned force fields. Recently, ConfGF (Shi et al., 2021) specifically tackled the conformer generation problem by estimating the gradient fields of inter-atomic distances (similar to force fields), where the 3D coordinates of the conformer can be generated through steps of Langevin dynamics achieving state-of-the-art performance. While our model also tries to estimate the gradient field, we instead employ an unrolled optimization architecture (Domke, 2012; Liu et al., 2018) aiming to model the entire optimization process instead of a single force field optimization step. Such one-pass end-to-end design is proved to be more effective empirically as shown in the experiment section. In addition, our approach also enjoys a much faster run time.

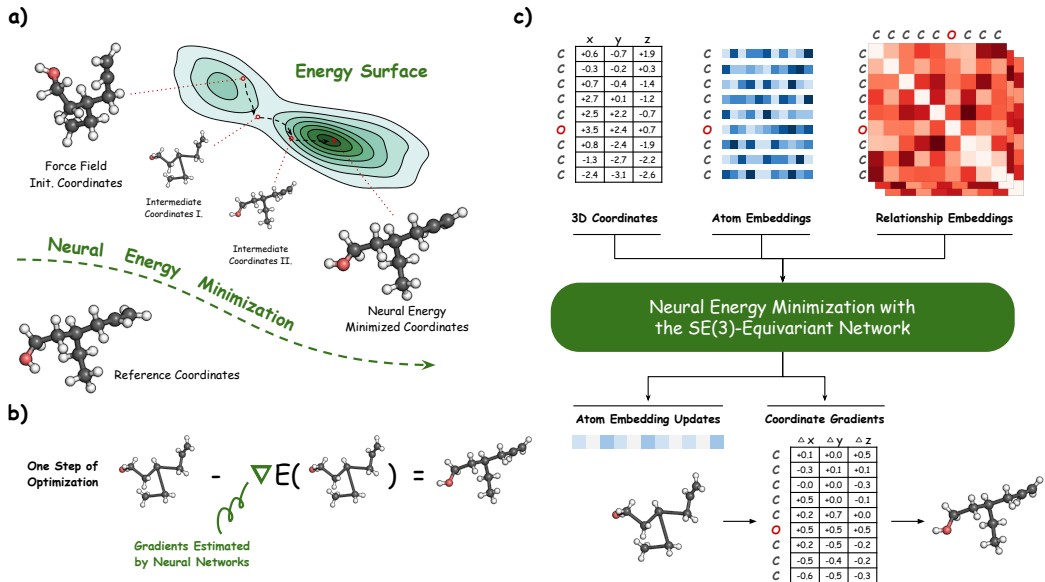

**Figure 1: Method Overview. a)** Our energy-inspired neural optimization formulation aims to optimize the initial conformer of a molecule towards the most stable and energy-minimized state, following an energy gradient field approximated by a neural network. **b)** To minimize the conformational energy, we can perform gradient descent for the 3D coordinates along the energy landscape, where the gradient is estimated by the model. **c)** To model the gradient fields, the neural network $\phi$ considers the current 3D conformation, the atom representations, and the relationship embeddings. After layers of non-linear transformations and SE(3)-equivariant operations, the model returns the gradient updates for the 3D coordinate along with the updated atom representations.

## 2.2 SE(3)-EQUIVARIANT NEURAL NETWORKS

Since 3D roto-translation symmetry is a common property for many physical systems, various methods have been proposed to model such symmetry in problems like quantum mechanics (Ramakrishnan et al., 2014), molecular design (Simm et al., 2020), particle simulations (Thomas et al., 2018), and point clouds (Uy et al., 2019). The ability to capture the spatial symmetry makes these models quite suitable for conformer optimization. In general, there are two ways to introduce the equivariance into a model. One thread of research relies on building models with irreducible representations and spherical harmonics (Thomas et al., 2018; Anderson et al., 2019; Finzi et al., 2020; Fuchs et al., 2020; Batzner et al., 2021). Different from this line of work, our approach aims to avoid the complexity of such mathematical tools, considering the higher-order features in these more theoretical framework might not be practical for our purpose. The other line of research relies on vector-based methods, where equivariant filters are constructed by taking the gradient of invariant filters (Schütt et al., 2021) or designed (Satorras et al., 2021) based on the intuition of message passing (Gilmer et al., 2017). By explicitly connecting an underlying energy function with an SE(3)-equivariant neural network in our framework, we can not only reinterpret some of the existing models but also derive new variants that aligns the model parametrization with an underlying physical assumption as shown in Section 3.3.

## 3 METHODS

### 3.1 PROBLEM DEFINITION

We represent each molecule as a graph $\mathcal{G}_{\text{mol}}$ and use a set of 3D coordinates $\boldsymbol{X}$ to denote the conformer. The molecular graph is defined as $\mathcal{G}_{\text{mol}} = \langle \mathcal{V}, \mathcal{E} \rangle$ where $\mathcal{V} = \{\boldsymbol{v}_i\}$ is the set of nodes representing atom features, and $\mathcal{E} = \{\boldsymbol{e}_i\}$ is the set of edges representing the relationship between atoms: including covalent interactions and distance of their shortest path. $\boldsymbol{X}$ consists of the 3D coordinate $\{\boldsymbol{x}_i | \boldsymbol{x}_i \in \mathbb{R}^3\}$ of each atom. Given a molecular graph $\mathcal{G}_{\text{mol}}$ and an initial conformer $\boldsymbol{X}^0$, we want to learn an optimization model $\phi$ that refines the initial conformation towards an energetically preferred state $\boldsymbol{X}^*$ as $\boldsymbol{X}^* = \arg\min_{\boldsymbol{X}} E(\boldsymbol{X} | \mathcal{G}_{\text{mol}})$ where $E$ is some energy surface of nature.

## 3.2 Conformational Energy Minimization with a Neural Network

Molecular conformation represents a 3D arrangement of the atoms and shares a close relationship with the conformational energy, as each equilibrium of a molecular conformation can be considered as a local minimum of the potential energy surface (Axelrod & Gomez-Bombarelli, 2020). One could therefore find these stable conformations through gradient descent as a process of energy minimization. Specifically, given a particular energy function $E$, we can optimize the conformation $\boldsymbol{X}$ by updating the 3D Cartesian coordinates with gradient descent such that $E$ can be minimized:

$$\boldsymbol{X}^{t+1} = \boldsymbol{X}^t - \gamma \nabla_{\boldsymbol{X}} E(\boldsymbol{X}^t | \mathcal{G}_{\mathrm{mol}}) \tag{1}$$

where $\gamma$ is the learning rate, and we will observe $\boldsymbol{X}^* = \boldsymbol{X}^* - \gamma \nabla_{\boldsymbol{X}} E(\boldsymbol{X}^* | \mathcal{G}_{\mathrm{mol}})$ for an energetically stable conformer. Therefore, we can easily find a stable molecular conformation from an initial conformation by applying the gradient descent update iteratively *if* the energy function $E(\boldsymbol{X} | \mathcal{G}_{\mathrm{mol}})$ is fully characterized and differentiable with respect to $\boldsymbol{X}$. Unfortunately, this is often not the case.

Instead of hand-designing the energy function and its gradient, we can take advantage of the approximation power of a neural network and use it to learn the gradient field from the data directly. Once the network is trained, we can then unroll the optimization (Domke, 2012; Liu et al., 2018) with the model and perform a neural version of the above optimization where the gradient term $\nabla E$ in Equation 1 is not derived from an explicit energy function but estimated by a neural network.

More concretely, we parameterize the energy surface as a function of the molecular graph $\mathcal{G} = \langle \mathcal{V}, \mathcal{E} \rangle$ and the 3D conformer $\boldsymbol{X}$. We can consider $\boldsymbol{X}$ as an external 3-dimensional representation of the molecule, while $\mathcal{V}$ and $\mathcal{E}$ represent an internal high dimensional representation of the molecule. These internal embeddings $\mathcal{V}$ and $\mathcal{E}$ not only encode the initial atom/bond information but also embed information about their neighborhood related to the evolving conformation $\boldsymbol{X}$. For each step, we want to minimize $E(\boldsymbol{X}, \mathcal{V}, \mathcal{E})$ with the following gradient updates:

$$\boldsymbol{x}_i^{t+1} = \boldsymbol{x}_i^t + \phi_{\boldsymbol{x}}(\mathcal{G}^t, \mathcal{E}^t) \quad \boldsymbol{v}_i^{t+1} = \boldsymbol{v}_i^t + \phi_{\boldsymbol{v}}(\mathcal{G}^t, \mathcal{E}^t) \quad \boldsymbol{e}_{ij}^{t+1} = \boldsymbol{e}_{ij}^t + \phi_{\boldsymbol{e}}(\mathcal{G}^t, \mathcal{E}^t)$$

for all $\boldsymbol{x}_i \in \boldsymbol{X}$, $\boldsymbol{v}_i \in \mathcal{V}$, and $\boldsymbol{e}_{ij} \in \mathcal{E}$, where $\phi$ are neural networks estimating their respective gradient field, and the exact parametrization of the neural network will depend on the energy function as described in the following section. In this work, we view $\boldsymbol{e}_{ij}$ as a constant representation for simplicity, and therefore, it is not updated in our models.

## 3.3 Recipe for deriving energy-inspired SE(3)-equivariant models

From the perspective of implicit energy minimization, we can derive various models by making different assumptions of the target energy function, resulting in a principle architectural design that aligns with the underlying physical assumption. In Appendix A, we show that a model derived from an SE(3)-invariant energy function (as they should be) always enjoys the SE(3)-equivariance naturally. This section instantiates three variants of SE(3)-equivariant networks as examples.

**Basic Two-Atom Model** We start with a simple energy formulation where the conformation energy considers atom pairs independently:

$$E(\boldsymbol{X}, \mathcal{V}, \mathcal{E}) = \sum_{i,j \in \mathcal{V}, i \neq j} u(d_{ij}^2, \boldsymbol{v}_i, \boldsymbol{v}_j, \boldsymbol{e}_{ij}) \tag{2}$$

where $d_{ij} = \|\boldsymbol{x}_i - \boldsymbol{x}_j\|$ is the euclidean distance between two atoms $i$ and $j$ while $u(\cdot)$ is some unknown the potential energy function in nature. Following the formulation, we can then derive the gradient field of atom coordinates:

$$-\frac{\partial E(\boldsymbol{X}, \mathcal{V}, \mathcal{E})}{\partial \boldsymbol{x}_i} = -\sum_{j \in \mathcal{V}, i \neq j} \frac{\partial u(d_{ij}^2, \boldsymbol{v}_i, \boldsymbol{v}_j, \boldsymbol{e}_{ij})}{\partial \boldsymbol{x}_i} = -\sum_{j \in \mathcal{V}, i \neq j} \frac{\partial d_{ij}^2}{\partial \boldsymbol{x}_i} \frac{\partial u(d_{ij}^2, \boldsymbol{v}_i, \boldsymbol{v}_j, \boldsymbol{e}_{ij})}{\partial d_{ij}^2}$$

$$= -\sum_{j \in \mathcal{V}, i \neq j} 2(\boldsymbol{x}_i - \boldsymbol{x}_j) \frac{\partial u(d_{ij}^2, \boldsymbol{v}_i, \boldsymbol{v}_j, \boldsymbol{e}_{ij})}{\partial d_{ij}^2} \approx \sum_{j \in \mathcal{V}, i \neq j} (\boldsymbol{x}_i - \boldsymbol{x}_j) f_x(d_{ij}^2, \boldsymbol{v}_i, \boldsymbol{v}_j, \boldsymbol{e}_{ij})$$

where $f_x$ is a Transformer-based neural network (Vaswani et al., 2017) approximating the gradient field ($\frac{\partial u}{\partial d_{ij}^2}$) with a constant scalar. The optimization step for a two-atom model therefore becomes:

$$\boldsymbol{x}_i^{t+1} = \boldsymbol{x}_i^t + \sum_{j \in \mathcal{V}, i \neq j} (\boldsymbol{x}_i^t - \boldsymbol{x}_j^t) f_x(d_{ij}^t, \boldsymbol{v}_i^t, \boldsymbol{v}_j^t, \boldsymbol{e}_{ij}^t) \tag{3}$$

and since $f_x$ is SE(3)-invariant and $(\boldsymbol{x}_i^t - \boldsymbol{x}_j^t)$ is rotational equivariant, the optimization step Equation 3 observes SE(3)-equivariance (proof in Appendix A). The equivariance here is consistent with the invariant nature of the energy function, as the gradient field should rotate with the input coordinate. Similarly, we can derive the following update for the internal atom representation:

$$\boldsymbol{v}_i^{t+1} = \boldsymbol{v}_i^t + \sum_{j \in \mathcal{V}, i \neq j} f_{\boldsymbol{v}}(d_{ij}^t, \boldsymbol{v}_i^t, \boldsymbol{v}_j^t, \boldsymbol{e}_{ij}^t). \tag{4}$$

**High-Order / Three-Atom Model**  While we start with an energy function that only considers atom pairs, the energy function can also be extended to consider atom triplets allowing us to capture the potential energy related to different bond angles:

$$E(\boldsymbol{X}, \mathcal{V}, \mathcal{E}) = \sum_{i,j,k \in \mathcal{V}, i \neq j, i \neq k} u(d_{ij}^2, d_{ik}^2, \langle \boldsymbol{r}_{ij}, \boldsymbol{r}_{ik} \rangle, \mathcal{V}_{ijk}, \mathcal{E}_{ijk})$$

where $\boldsymbol{r}_{ij} = \boldsymbol{x}_i - \boldsymbol{x}_j$ and $\langle \cdot, \cdot \rangle$ denotes the inner product of two vectors capturing bond angle $\angle_{jik}$.

The dependency on internal atom and edge embeddings is also extended to atom triplets $\mathcal{V}_{ijk} = \{\boldsymbol{v}_i, \boldsymbol{v}_j, \boldsymbol{v}_k\}$ and their corresponding relationship representations $\mathcal{E}_{ijk} = \{\boldsymbol{e}_{ij}, \boldsymbol{e}_{ik}, \boldsymbol{e}_{jk}\}$. Following the derivation in Appendix B, we observe the following updates for the three-atom model:

$$\begin{aligned}
\boldsymbol{x}_i^{t+1} = \boldsymbol{x}_i^t + \sum_{j,k \in \mathcal{V}, i \neq j, i \neq k} &[(\boldsymbol{x}_i^t - \boldsymbol{x}_j^t) f_x(d_{ij}^t, d_{ik}^t, \langle \boldsymbol{r}_{ij}^t, \boldsymbol{r}_{ik}^t \rangle, \mathcal{V}_{ijk}^t, \mathcal{E}_{ijk}^t) \\
&+ (\boldsymbol{x}_i^t - \boldsymbol{x}_k^t) g_x(d_{ij}^t, d_{ik}^t, \langle \boldsymbol{r}_{ij}^t, \boldsymbol{r}_{ik}^t \rangle, \mathcal{V}_{ijk}^t, \mathcal{E}_{ijk}^t)]
\end{aligned} \tag{5}$$

where SE(3)-equivariance is also achieved. Similarly, we have the following update function for the internal atom representation: $\boldsymbol{v}_i^{t+1} = \boldsymbol{v}_i^t + \sum_{j,k \in \mathcal{V}, i \neq j, i \neq k} f_{\boldsymbol{v}}(d_{ij}^t, d_{ik}^t, \langle \boldsymbol{r}_{ij}^t, \boldsymbol{r}_{ik}^t \rangle, \mathcal{V}_{ijk}^t, \mathcal{E}_{ijk}^t)$.

While we only extend the framework to atom triplets, one could extend the framework to an even higher-order model to capture the energy contribution for geometric structures like torsional angles.

**Geometry of the Internal Representation**  In the basic two-atom model, we derive a direct $\boldsymbol{v}$ update formula as Equation 4. However, we can also view $\boldsymbol{v}$ as an internal spatial organization of the atoms, and therefore, the proximity ($\mathcal{P}$) between two $n$-dim vectors also contributes to the energy:

$$E(\boldsymbol{X}, \mathcal{V}, \mathcal{E}) = \sum_{i,j \in \mathcal{V}, j \neq i} u(d_{ij}^2, \boldsymbol{v}_i, \boldsymbol{v}_j, \mathcal{P}(\boldsymbol{v}_i, \boldsymbol{v}_j), \boldsymbol{e}_{ij})$$

where $\mathcal{P}$ is some distance metric for $\boldsymbol{v}_i$ and $\boldsymbol{v}_j$. For instance, using cosine similarity $\mathcal{P}(\boldsymbol{v}_i, \boldsymbol{v}_j) = \frac{\boldsymbol{v}_i^T \boldsymbol{v}_j}{\|\boldsymbol{v}_i\| \|\boldsymbol{v}_j\|}$ as the distance metric, we arrive at the following update by taking the gradient of $E$ w.r.t $\boldsymbol{v}_i$:

$$\boldsymbol{v}_i^{t+1} = \boldsymbol{v}_i^t + \sum_{j \in \mathcal{V}, i \neq j} [f_{\boldsymbol{v}}(d_{ij}^t, \boldsymbol{v}_i, \boldsymbol{v}_j, \mathcal{P}(\boldsymbol{v}_i, \boldsymbol{v}_j), \boldsymbol{e}_{ij}) + \frac{\boldsymbol{v}_j}{\|\boldsymbol{v}_j\|} f_{\mathcal{P}}(d_{ij}^t, \boldsymbol{v}_i, \boldsymbol{v}_j, \mathcal{P}(\boldsymbol{v}_i, \boldsymbol{v}_j))]. \tag{6}$$

**Reinterpretation for existing models**  Besides new variants, the idea of implicit energy minimization also allows us to reinterpret some existing models. For instance, in EGNN (Satorras et al., 2021), the authors propose the following updates for atom "representations" $\boldsymbol{v}$ and "coordinates" $\boldsymbol{x}$:

$$\boldsymbol{v}_i^{t+1} = \phi_h(\boldsymbol{v}_i^t, \sum_{j \in \mathcal{N}(i)} \phi_e(d_{ij}^2, \boldsymbol{v}_i^t, \boldsymbol{v}_j^t, \boldsymbol{e}_{ij}))$$

$$\boldsymbol{x}_i^{t+1} = \boldsymbol{x}_i^t + \sum_{j \in \mathcal{V}, i \neq j} (\boldsymbol{x}_i^t - \boldsymbol{x}_j^t) \phi_x(\phi_e(d_{ij}^2, \boldsymbol{v}_i^t, \boldsymbol{v}_j^t, \boldsymbol{e}_{ij})).$$

From the new perspective, these are equivalent to optimizing an conformational energy function that depends on both the atom representations and inter-atomic distances just like our basic Two-Atom model, i.e. $E = \sum_{i,j \in \mathcal{V}, i \neq j} u(d_{ij}^2, \boldsymbol{v}_i, \boldsymbol{v}_j, \boldsymbol{e}_{ij})$. On the other hand, the "gradient" for their atom representations are only aggregated from the corresponding neighbors (as message passing (Gilmer et al., 2017)) instead of all atoms (as ours). In addition, this formulation uses a learned function $\phi_h$ to update the representation $\boldsymbol{v}_i^t$ instead of simply adding the negative gradient. In the context of optimization, the operation can be interpreted as optimizing with a learned optimizer (Metz et al.,

2019) instead of performing simple gradient descent. Additional reinterpretations for existing models can be found in Appendix. C.

We believe the connection between the models and underlying energy functions represents a new perspective for SE(3)-equivariant models and expect the community to derive more interesting variants by designing different energy functions with more suitable assumptions for their applications.

# 4 EXPERIMENTS

To evaluate the proposed methods, we perform experiments in two separated settings:

**The optimization setting** where the goal for the model is to take a 3D conformer as input and produce a *single* optimized conformer with the most stable energy. Since our model is proposed to solve an optimization task, this setting is a direct measure of its performance against other SE(3)-equivariant models also capable of taking 3D Cartesian coordinates as input. Our results suggest that models derived from higher-order energy function can achieve better performance, and we believe such energy-inspired perspective would also help the community to develop more powerful extensions of SE(3)-equivariant models in the future.

**The generation setting** where the goal is to generate *multiple* conformers that capture a diverse set of relatively stable conformers. This setting has been studied recently, even outside the context of SE(3)-equivariant model (Mansimov et al., 2019; Simm & Hernandez-Lobato, 2020; Xu et al., 2020; Shi et al., 2021), and represents a more realistic application scenario for practitioners. Our experiment suggests that the unrolled optimization formulation can be easily extended for the generative setting. Compared to existing baselines, our models can generate a more diverse and accurate ensemble of conformers, demonstrating a realistic application scenario for the proposed formulation.

## 4.1 MOLECULAR CONFORMER OPTIMIZATION

**Data** We test molecular conformer optimization on the QM9 dataset with small molecules (up to 9 heavy atoms) (Ramakrishnan et al., 2014) as well as the GEOM-Drugs dataset with larger molecules (up to 91 heavy atoms) (Axelrod & Gomez-Bombarelli, 2020). For the QM9 dataset, only one reference conformer generated through DFT calculation is considered as the lowest-energy conformer. For the GEOM-Drugs dataset, since multiple low-energy conformers are given through a semi-empirical method and after realizing larger molecules could have multiple similarly optimal conformers, we select multiple conformers with high Boltzmann weights as reference conformers (see Appendix D for details). We randomly split the QM9 dataset into 123k, 5k, 5k, and the GEOM-Drugs dataset into 270K, 10K, and 10K for respective training, validation, and testing.

**Baselines** Since the goal for this experiment is to show the energy-derived model is indeed capable of **optimizing** an existing 3D conformer towards its most energy stable state, we compare our model with two state-of-the-art SE(3)-equivariant models: **EGNN** (Satorras et al., 2021) and **SE(3)-Transformer** (Fuchs et al., 2020) that can also take 3D Cartesian coordinates as input directly. For non-machine-learning method, we also include one classical approach **RDKit + MMFF** (Riniker & Landrum, 2015), where the predict conformer is estimated through Euclidean Distance Geometry (EDG) and further optimized with Merck Molecular Force Field (MMFF) (Halgren, 1996).

**Setup** Following Sec. 3.3, we include all three variants for this experiment: (1) **Ours-TwoAtom**, a basic two-atom model where we only consider pairwise atomic distances in the energy function. (2) **Ours-Ext$_v$**, which extends the two-atom model by considering the geometry of the internal representation $v$ in the energy function. (3) **Ours-ThreeAtom**, which further extends the two variants above by considering the atom triplets in the energy function. To train our models and other machine learning baselines, we optimize model parameters by minimizing the $L2$ loss between pairwise distance matrices of the generated conformer and its closest reference conformer (see Appendix E for more model details). Taking advantage of the SE(3)-equivariant property of the baseline and our models, we train all the models to take the **RDKit + MMFF** predicted conformers as input and allow the models to start with a reasonable estimation at a low computation cost. We perform same number of optimization step (=9) for all models and both datasets. The results for random initialization can also be found in Appendix H.1, and the results for effects of the number of optimization steps can be found in Appendix K.

| Dataset / Model | QM9 | | GEOM-Drugs | |
|---|---|---|---|---|
| | mean RMSD (↓) | median RMSD (↓) | mean RMSD (↓) | median RMSD (↓) |
| RDKit+MMFF | 0.3872 ± 0.0029 | 0.2756 ± 0.0075 | 1.7913 ± 0.0030 | 1.6433 ± 0.0097 |
| SE(3)-Tr. (Fuchs et al., 2020) | 0.2476 ± 0.0021 | 0.1657 ± 0.0022 | 1.0050 ± 0.0022 | 0.9139 ± 0.0041 |
| EGNN (Satorras et al., 2021) | 0.2101 ± 0.0009 | 0.1356 ± 0.0013 | 1.0405 ± 0.0018 | 0.9598 ± 0.0038 |
| Ours-TwoAtom | 0.1415 ± 0.0004 | 0.0534 ± 0.0002 | 0.8839 ± 0.0014 | 0.7733 ± 0.0026 |
| Ours-Ext$_v$ | 0.1383 ± 0.0005 | **0.0505** ± 0.0001 | 0.8691 ± 0.0015 | 0.7535 ± 0.0028 |
| Ours-ThreeAtom | **0.1374** ± 0.0004 | 0.0522 ± 0.0002 | **0.8567** ± 0.0014 | **0.7192** ± 0.0024 |

**Table 1:** Molecular conformer optimization for the QM9 dataset and the GEOM-Drugs dataset. The mean and median RMSDs (Unit Å) between reference and predicted conformers are reported. The confidence interval is calculated by inference with ten different initialization from RDKit+MMFF. (↓) denotes lower scores are better. More studies on model ablations and variations can be found in Appendix I.

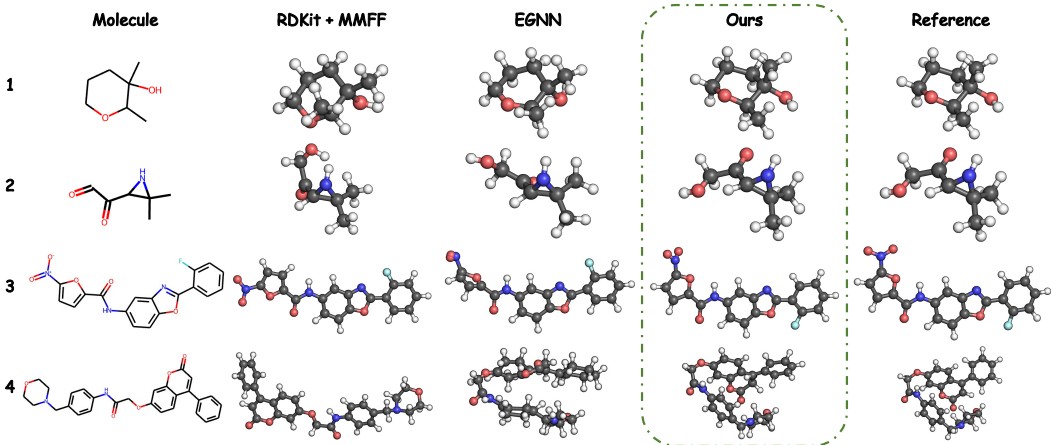

**Figure 2: Model-Predicted Conformers.** Molecule 1 & 2 are sampled from the QM9 dataset, and the larger molecule 3 & 4 are sampled from the GEOM-Drugs dataset. The reference conformer, as well as the initial conformations from RDKit+MMFF, are also shown. See Appendix L for more examples and failure cases.

**Results** Following Mansimov et al. (2019), we measure the difference between a predicted conformer and its reference by the Root-Mean-Square Deviation (RMSD) for all the heavy atoms. As shown in Table 1, all the optimization models can improve upon the initial conformation estimated by RDKit+MMFF, and new variants derived from the new framework also outperform the two state-of-the-art SE(3)-equivariant models with a clear margin. More interestingly, model variants derived with more complex and higher-order energy definitions also outperform the basic model that considers inter-atomic distance. In the GEOM-Drugs dataset, where we have large molecules with many rotatable bonds, we hypothesize the higher-order model can outperform the basic two-atom model because the corresponding higher-order energy function can better approximate the actual potential energy surface by considering the bond-angle explicitly. Since we only consider three variants in this work and they all have shown great performance, we look forward to other more powerful variants the community can develop with such perspective.

In Figure 2, examples of model-predicted conformers are visualized along with their initial conformers from RDKit+MMFF and the reference conformers in their optimal energy state. The visualization shows that our models can produce conformers that are not only realistic but also energetically stable. With no intentional design, our model-predicted conformers can also capture the structure of the basic components quite well (e.g., benzene forms a ring in the same plane). For larger molecules where the initial conformers (RDKit+MMFF) are not folded/rotated in the most stable fashion, our model can also optimize the conformers towards the stable state as reference conformers.

To measure the quality of the optimized conformers beyond the RMSD metric, we evaluate two downstream applications with the QM9 test set conformers predicted by different methods. In the first study, we compare the HOMO and LUMO energy (Smith et al., 2020) of a molecule when calculated

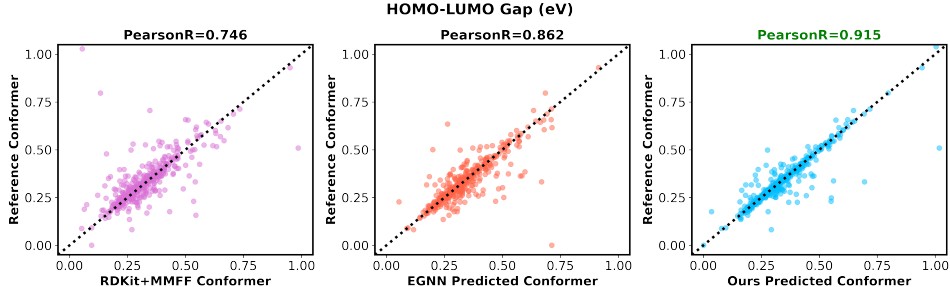

**Figure 3:** HOMO-LUMO gap calculated with reference conformers vs. ones calculated with the predicted conformers. The black dotted line represents the identical values. Four hundred molecules are sampled.

| Task / Conformer | $\alpha$ bohr$^3$ | $\Delta\epsilon$ meV | $\epsilon_{\text{HOMO}}$ meV | $\epsilon_{\text{LUMO}}$ meV | $\mu$ D | $C_\nu$ cal/mol K |
|---|---|---|---|---|---|---|
| Reference | 0.235 | 63 | 41 | 34 | 0.033 | 0.033 |
| RDKit+MMFF | 0.277 | 112 | 83 | 76 | 0.394 | 0.110 |
| EGNN-Predicted | 0.278 | 115 | 91 | 70 | 0.328 | 0.111 |
| Ours-Predicted | **0.254** | **103** | **82** | **64** | **0.274** | **0.099** |

**Table 2:** SchNet (Schütt et al., 2017) model performance (MAE) on QM9 property prediction when trained and evaluated with predicted conformers instead of the reference conformers. The best performance is **bolded**.

with the reference conformers vs. the model-predicted conformers. As shown in Figure 3, we find the HOMO-LUMO gap calculated from our model-predicted conformer match well with the one from reference, while the correlation between the reference and force field optimized conformers is much lower. In the second study, we train a standard neural network model, SchNet (Schütt et al., 2017) for the QM9 benchmark (Ramakrishnan et al., 2014) with the reference or predicted conformers. As shown in Table 2, the model trained with our predicted conformers can outperform the ones trained with other predicted conformers. The result indicates better inductive bias in our predicted 3D structures compared to the ones estimated using other algorithms. Even though there is still a performance gap when comparing results with the model trained with the reference conformers, we believe the model-predicted conformer could still be helpful for datasets where no references are not available. More experiment details can be found in Appendix F.

## 4.2 MOLECULAR CONFORMER GENERATION

**Data** We test molecular conformer generation on both GEOM-QM9 and GEOM-Drugs datasets (Axelrod & Gomez-Bombarelli, 2020) following the same data split as Shi et al. (2021) including 40k training molecules and 200 test molecules for both datasets (see Appendix D for details).

**Baselines** We compare our model against an EDG-based approach **RDKit** (Riniker & Landrum, 2015) as well as four deep generative models specifically studied for the generation setting: **CVGAE** (Mansimov et al., 2019) is a conditional VAE that directly generates atom coordinates based on molecular graphs. **GraphDG** (Simm & Hernandez-Lobato, 2020) and **CGCF** (Xu et al., 2020) are VAE- and flow-based method that only generate the pairwise distances for 3D conformer conversion. **ConfGF** (Shi et al., 2021) is a recently-published state-of-the-art model that learns the gradient field for molecular conformations and perform generation through iterative Langevin sampling.

**Setup** In order to test our optimization formulation for the generative setting, we adopt the same two-atom model from the optimization setting here with *the same hyper-parameters*. However, instead of training the model to optimize a single initialization towards one reference, we train the model to optimize $K$ initialization such that these $K$ optimized conformers can be exactly matched to $K$ different sampled references. More concretely, we optimize the model parameters against an optimal-transport-like loss similar to other generative models (Genevay et al., 2018):

$$L_{OT} = \min_{\pi \in \Gamma} \sum_{i,j} \pi_{i,j} \mathbf{C}(\boldsymbol{X}_i^*, \boldsymbol{X}_j) \tag{7}$$

| Dataset | GEOM-QM9 | | | | | | GEOM-Drugs | | | | | |
|---|---|---|---|---|---|---|---|---|---|---|---|---|
| Metric | COV(%) (↑) | | MIS(%) (↓) | | MAT(Å) (↓) | | COV(%) (↑) | | MIS(%) (↓) | | MAT(Å) (↓) | |
| | Mean | Median | Mean | Median | Mean | Median | Mean | Median | Mean | Median | Mean | Median |
| RDKit | 83.26 | 90.78 | 8.13 | 1.00 | 0.3447 | 0.2935 | 60.91 | 65.70 | 27.95 | 12.07 | 1.2026 | 1.1252 |
| CVGAE | 0.09 | 0.00 | - | - | 1.6713 | 1.6088 | 0.00 | 0.00 | - | - | 3.0702 | 2.9937 |
| GraphDG | 73.33 | 84.21 | 56.09 | 64.66 | 0.4245 | 0.3973 | 8.27 | 0.00 | 97.92 | 100.00 | 1.9722 | 1.9845 |
| CGCF | 78.05 | 82.48 | 63.51 | 64.66 | 0.4219 | 0.3900 | 53.96 | 57.06 | 78.32 | 86.28 | 1.2487 | 1.2247 |
| ConfGF | 88.49 | **94.13** | 53.56 | 56.59 | 0.2673 | 0.2685 | 62.15 | 70.93 | 76.58 | 84.48 | 1.1629 | 1.1596 |
| Ours-Random | **88.83** | 93.18 | 30.21 | 30.74 | 0.3778 | 0.3736 | **76.50** | **83.78** | 31.40 | 23.03 | **1.0694** | 1.0583 |
| Ours-RDKit | 86.68 | 91.34 | **5.46** | **0.00** | **0.2667** | **0.2125** | 67.72 | 75.30 | **21.52** | **2.66** | 1.0739 | **1.0372** |

**Table 3:** Molecular conformer generation for the GEOM-QM9 and GEOM-Drugs dataset. **COV**erage score reports the percentage of reference conformers that are produced by the predicted ensemble. **MIS**matching score reports the percentage of generated conformers that can not be matched with any reference conformer. **MAT**ching score reports the minimum RMSD between a generated conformer and the references. (↓)/(↑) denotes a metric for which lower/higher scores are better. The best performance is **bolded**.

where $\mathbf{C}(\boldsymbol{X}_i^*, \boldsymbol{X}_j)$ is the same $L2$ loss between the distance matrices of the reference ($\boldsymbol{X}_i^*$) and generated ($\boldsymbol{X}_j$) conformer, while the optimal transport plan $\pi \in \{0,1\}^{K \times K}$ is realized as a minimum weight bipartite graph matching and $\Gamma$ is the set of all permutations. For the initial conformer, we again take advantage of the SE(3)-equivariance of our method and evaluate the generation performance with initialization from either a random set of coordinates (i.e., **Ours-Random** model) or a rough RDKit estimate (i.e., **Ours-RDKit** model, whose training inputs are perturbed references).

**Results** To evaluate the diversity, precision, and structural quality of generated conformers, we adopt the corresponding **COV**erage, **MIS**matching, and **MAT**ching scores following Xu et al. (2020) and Shi et al. (2021) as defined in Appendix G. As shown in Table. 3, our models achieve competitive performance in all three metrics, especially for the more challenging GEOM-Drugs dataset where we outperform existing machine learning methods by a clear margin. These results suggest that our optimization formulation can be easily extended to the generation setting for a diverse and accurate ensemble of conformers. Ablation study in Appendix J also suggests the improvement is mainly contributed by the SE(3)-equivariant setup and the optimal transport loss. Since no Langevin sampling is required in our method, the inference time (<1s per molecule) for our method is also much faster than ConfGF (∼170s per molecule). We also noted the performance difference between random initialization and RDKit initialization also represents a classical trade-off between diversity and accuracy, where a RDKit initialization can achieve better accuracy (MAT / MIS) by providing a more accurate starting point, while a random initialization can achieve better coverage (COV) by sufficient sampling of the space. We left more interesting initialization (e.g. mixture of the two) for future work, and please consult Appendix G for details of the implementations.

## 5 CONCLUSION AND DISCUSSION

This paper proposes a new formulation for molecular conformer prediction by parametrizing an SE(3)-equivariant network to model the gradient field of the conformational energy landscape. By connecting an SE(3)-equivariant model with an underlying energy function, we also found a new perspective to principally derive new variants of SE(3)-equivariant models that align with an explicit set of underlying assumptions for the physical system. Through an extensive set of experiments, we show that the proposed method is capable of *optimizing* a given conformer towards its most energetic-stable state and *generating* an ensemble of diverse and accurate conformers efficiently. We believe such energy-inspired perspective also represents a new direction for thinking about SE(3)-equivariant model and expect the community to derive more interesting variants with energy assumptions that are more suitable for their applications.

For future work, our interest also goes beyond conformations of small molecules, as we can further scale up the system to model larger bio-molecular systems such as protein-ligand complexes. In addition, it would be interesting to explore models like DEQ (Bai et al., 2019) where the equilibrium is modeled directly. Since our training objective is blind to the chirality, our system will need to infer the chirality post model generation, and it remains an interesting question that whether we can build the system with chirality built-in (Pattanaik et al., 2020). Last but not least, with the ability to quickly generate high-quality ensembles of molecular conformers, we are also interested in molecular machine learning with conformer ensembles (Axelrod & Gómez-Bombarelli, 2020).

**Reproducibility Statements**    The model implementation, experimental data and model checkpoints can be found here: `https://github.com/guanjq/confopt_official`

**Acknowledgement**    We thank all the reviewers for their feedbacks through out the review cycles of the manuscript. We also thank Yuanyi Zhong and Yunan Luo for many helpful discussions. This work was supported by U.S. National Science Foundation under grant no. 2019897 and U.S. Department of Energy award DE-SC0018420. J.P. acknowledges the support from the Sloan Research Fellowship and the NSF CAREER Award.

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

## A   PROOF OF SE(3)-EQUIVARIANCE

### A.1   PROOF OF SE(3)-EQUIVARIANCE FOR THE TWO-ATOM MODEL

One of the crucial challenges for modeling molecular conformers is to capture the 3D roto-translation symmetry, as rotating or moving the conformer in the 3D space on its own would not change the conformational energy. Therefore, to capture such symmetry for better generalization (Köhler et al., 2019), our model is required to change its output accordingly if the input conformer $\boldsymbol{X}$ undergoes a transformation of rotation or/and translation in the 3D space. More concretely, the model needs to satisfy SE(3)-equivariance:

$$\phi(T(\boldsymbol{X})) = T(\phi(\boldsymbol{X}))$$

where $T : \boldsymbol{X} \rightarrow \hat{\boldsymbol{X}}$ is a transformation on $\boldsymbol{X}$ for the SE(3) group. Specifically, for each 3D coordinate $\boldsymbol{x}$, we have $T(\boldsymbol{x}) = \boldsymbol{A}\boldsymbol{x} + \boldsymbol{b}$ where $\boldsymbol{A} \in \mathbb{R}^{3 \times 3}$ is the rotation matrix and $\boldsymbol{b} \in \mathbb{R}^3$ is the translation vector. In the rest of the section, we simply denote $\hat{\boldsymbol{x}}$ as $T(\boldsymbol{x})$.

Now that we have defined SE(3)-equivariance, we want to prove that the following coordinate update in Eq. equation 4 satisfies SE(3)-equivariant:

$$\boldsymbol{x}_i^{t+1} = \phi(\boldsymbol{x}_i^t) = \boldsymbol{x}_i^t + \sum_{j \in \mathcal{V}, i \neq j} (\boldsymbol{x}_i^t - \boldsymbol{x}_j^t) f_x(d_{ij}^t, \boldsymbol{v}_i^t, \boldsymbol{v}_j^t, \boldsymbol{e}_{ij}^t)$$

and we will do so by first showing that $f_x(\cdot)$ is SE(3)-invariant. To show $f_x(\cdot)$ is SE(3)-invariant, it is sufficient to show that its various inputs $d_{ij}$, $\boldsymbol{v}_i$, and $\boldsymbol{e}_{ij}$ are SE(3)-invariant. Intuitively, $d_{ij}$ should not change with the 3D roto-translation $T$, but formally we have:

$$\hat{d}_{ij}^2 = \|\hat{\boldsymbol{x}}_i - \hat{\boldsymbol{x}}_j\|^2 = \|(\boldsymbol{A}\boldsymbol{x}_i + \boldsymbol{b}) - (\boldsymbol{A}\boldsymbol{x}_j + \boldsymbol{b})\|^2 = \|\boldsymbol{A}\boldsymbol{x}_i - \boldsymbol{A}\boldsymbol{x}_j\|^2$$
$$= (\boldsymbol{x}_i - \boldsymbol{x}_j)^\top \boldsymbol{A}^\top \boldsymbol{A}(\boldsymbol{x}_i - \boldsymbol{x}_j) = (\boldsymbol{x}_i - \boldsymbol{x}_j)^\top \mathbf{I}(\boldsymbol{x}_i - \boldsymbol{x}_j) = \|\boldsymbol{x}_i - \boldsymbol{x}_j\|^2 = d_{ij}^2$$

Since the initialization and update of $\boldsymbol{v}$ in Equation 4 only depend on variables satisfying SE(3)-invariant, $\boldsymbol{v}$ is also SE(3)-invariant. Similarly, the relationship embedding $\boldsymbol{e}$ is also SE(3)-invariant. Therefore, function $f_x$ satisfies SE(3)-invariant.

Now, if we look at the update for $\phi(T(\boldsymbol{x}^t))$ or $\phi(\hat{\boldsymbol{x}}^t)$, we have:

$$\phi(T(\boldsymbol{x}^t)) = \hat{\boldsymbol{x}}^t + \sum_{j \in \mathcal{V}, i \neq j} (\hat{\boldsymbol{x}}_i^t - \hat{\boldsymbol{x}}_j^t) \hat{f}_x(\cdot)$$
$$= \hat{\boldsymbol{x}}^t + \sum_{j \in \mathcal{V}, i \neq j} (\hat{\boldsymbol{x}}_i^t - \hat{\boldsymbol{x}}_j^t) f_x(\cdot)$$
$$= (\boldsymbol{A}\boldsymbol{x}^t + \boldsymbol{b}) + \sum_{j \in \mathcal{V}, i \neq j} \left((\boldsymbol{A}\boldsymbol{x}_i^t + \boldsymbol{b}) - (\boldsymbol{A}\boldsymbol{x}_j^t + \boldsymbol{b})\right) f_x(\cdot)$$
$$= (\boldsymbol{A}\boldsymbol{x}^t + \boldsymbol{b}) + \sum_{j \in \mathcal{V}, i \neq j} \boldsymbol{A}(\boldsymbol{x}_i^t - \boldsymbol{x}_j^t) f_x(\cdot)$$
$$= \boldsymbol{A}\left(\boldsymbol{x}^t + \sum_{j \in \mathcal{V}, i \neq j} (\boldsymbol{x}_i^t - \boldsymbol{x}_j^t) f_x(d_{ij}^t, \boldsymbol{v}_i^t, \boldsymbol{v}_j^t, \boldsymbol{e}_{ij}^t)\right) + \boldsymbol{b}$$
$$= \boldsymbol{A}\phi(\boldsymbol{x}^t) + \boldsymbol{b}$$
$$= T(\phi(\boldsymbol{x}^t))$$

Thus, the two-atom model update $\phi$ is SE(3)-equivariant.

### A.2   PROOF OF SE(3)-EQUIVARIANT FOR THE GENERAL CASE

For the more general case, we have the following equivariance theorem:

**Theorem 1.** *Let $G$ be an SE(3)-group which acts on $\mathbb{R}^{n \times 3}$. If the potential function $\Phi : \mathbb{R}^{n \times 3} \rightarrow \mathbb{R}$ is a G-invariant function (i.e. the assumed energy function $E$ shown in Equation 1 is G-invariant), then the gradient vector field $\nabla_x \Phi$ will be G-equivariant.*

*Proof.* For a rotation transformation $\forall g \in G$ with a rotation matrix $R_g$, we have the following if $\Phi$ is SE(3)-invariant: $\nabla_x(R_g \circ \Phi) = \nabla_x \Phi$. Now, according to the chain rule, we also have the following: $\nabla_x(R_g \circ \Phi) = (\nabla_x R_g x) \circ (\nabla_{R_g x} \Phi) = R_g^T \circ (\nabla_{R_g x} \Phi)$. Now putting the two equations together, and multiply the orthogonal matrix $R_g$ on both sides, we have: $R_g \circ \nabla_x \Phi(x) = R_g \circ R_g^T \circ \nabla_{R_g x} \Phi(R_g x) = \nabla_{R_g x} \Phi(R_g x)$. Similarly, for a transnational transformation $g(x) = x + t$, one could show that $\nabla_{x+t} \Phi(x + t) = \nabla_x \Phi(x)$ if $\Phi$ is SE(3)-invariant. The combination of these achieve the SE(3)-equivariant of the update formula:

$$(R_g x + t) - \nabla_{R_g x + t} \Phi(R_g x + t) = R_g(x - \nabla_x \Phi(x)) + t$$

i.e. $f \circ T_g = T_g \circ f$, where $f$ is our neural network and $T_g$ is the SE(3) transformation. $\qquad \square$

This shows that if the constructed energy function $E$ is SE(3)-invariant as it should be, it is sufficient to guarantee that the gradient field $\nabla E$ (and estimated neural network) are SE(3)-equivariant. The theorem can be generalized to any group $G$ utilizing the Riemannian geometry. We refer Wasserman (1969) and Katsman et al. (2021) to readers of interest.

## B   DERIVATION OF THE THREE-ATOM MODEL

For the three-atom model, we have the following energy formulation:

$$E(\boldsymbol{X}, \mathcal{V}, \mathcal{E}) = \sum_{i,j,k \in \mathcal{V}, i \neq j, i \neq k} u(d_{ij}^2, d_{ik}^2, \langle \boldsymbol{r}_{ij}, \boldsymbol{r}_{ik} \rangle, \mathcal{V}_{ijk}, \mathcal{E}_{ijk})$$

and following the energy function, we can derive the following gradient update w.r.t. $\boldsymbol{x}$:

$$
\begin{aligned}
-\frac{\partial E(\boldsymbol{X}, \mathcal{V}, \mathcal{E})}{\partial \boldsymbol{x}_i} &= - \sum_{i,j,k \in \mathcal{V}, i \neq j, i \neq k} \frac{\partial u(d_{ij}^2, d_{ik}^2, \langle \boldsymbol{r}_{ij}, \boldsymbol{r}_{ik} \rangle, \mathcal{V}_{ijk}, \mathcal{E}_{ijk})}{\partial \boldsymbol{x}_i} \\
&= - \sum_{i,j,k \in \mathcal{V}, i \neq j, i \neq k} [\frac{\partial d_{ij}^2}{\partial \boldsymbol{x}_i} \frac{\partial u(\cdot)}{\partial d_{ij}^2} + \frac{\partial d_{ik}^2}{\partial \boldsymbol{x}_i} \frac{\partial u(\cdot)}{\partial d_{ik}^2} + \frac{\partial \langle \boldsymbol{r}_{ij}, \boldsymbol{r}_{ik} \rangle}{\partial \boldsymbol{x}_i} \frac{\partial u(\cdot)}{\partial \langle \boldsymbol{r}_{ij}, \boldsymbol{r}_{ik} \rangle}] \\
&= - \sum_{i,j,k \in \mathcal{V}, i \neq j, i \neq k} [(\boldsymbol{x}_i - \boldsymbol{x}_j) \frac{2 \partial u(\cdot)}{\partial d_{ij}^2} + (\boldsymbol{x}_i - \boldsymbol{x}_j) \frac{2 \partial u(\cdot)}{\partial d_{ik}^2} + (2\boldsymbol{x}_i - \boldsymbol{x}_j - \boldsymbol{x}_k) \frac{\partial u(\cdot)}{\partial \langle \boldsymbol{r}_{ij}, \boldsymbol{r}_{ik} \rangle}] \\
&= - \sum_{i,j,k \in \mathcal{V}, i \neq j, i \neq k} [(\boldsymbol{x}_i - \boldsymbol{x}_j)(\frac{2 \partial u(\cdot)}{\partial d_{ij}^2} + \frac{\partial u(\cdot)}{\partial \langle \boldsymbol{r}_{ij}, \boldsymbol{r}_{ik} \rangle}) + (\boldsymbol{x}_i - \boldsymbol{x}_j)(\frac{2 \partial u(\cdot)}{\partial d_{ik}^2} + \frac{\partial u(\cdot)}{\partial \langle \boldsymbol{r}_{ij}, \boldsymbol{r}_{ik} \rangle})] \\
&\approx \sum_{i,j,k \in \mathcal{V}, i \neq j, i \neq k} [(\boldsymbol{x}_i - \boldsymbol{x}_j) f_x(\cdot) + (\boldsymbol{x}_i - \boldsymbol{x}_j) g_x(\cdot)]
\end{aligned}
$$

and therefore, we have the following $\boldsymbol{x}$ updates for the three-atom models:

$$
\begin{aligned}
\boldsymbol{x}_i^{t+1} = \boldsymbol{x}_i^t + \sum_{j,k \in \mathcal{V}, i \neq j, i \neq k} &[(\boldsymbol{x}_i^t - \boldsymbol{x}_j^t) f_x(d_{ij}^t, d_{ik}^t, \langle \boldsymbol{r}_{ij}^t, \boldsymbol{r}_{ik}^t \rangle, \mathcal{V}_{ijk}^t, \mathcal{E}_{ijk}^t) \\
&+ (\boldsymbol{x}_i^t - \boldsymbol{x}_k^t) g_x(d_{ij}^t, d_{ik}^t, \langle \boldsymbol{r}_{ij}^t, \boldsymbol{r}_{ik}^t \rangle, \mathcal{V}_{ijk}^t, \mathcal{E}_{ijk}^t)]
\end{aligned}
$$

## C   REINTERPRETATIONS OF EXISTING MODELS

### C.1   EQUIVARIANT BOLTZMANN GENERATOR

Equivariant Boltzmann Generator (Köhler et al., 2019) introduces an E(n)-equivariant update for $\boldsymbol{x}_i$ in the following form:

$$\boldsymbol{x}_i^{t+1} = \boldsymbol{x}_i^t + \sum_{j \in \mathcal{V}, i \neq j} (\boldsymbol{x}_i^t - \boldsymbol{x}_j^t) \phi(d_{ij})$$

where $\phi$ is a Multilayer Perceptron (MLP). It explicitly optimizes the energy function $E = \sum_{i,j \in \mathcal{V}}$, which can be interpreted as a simplified case of the energy function we proposed in Equation 2 since it only concerns about the distance between two nodes and disregards the information associated with the nodes and edges.

## C.2 TFN and SE(3)-Transformer

TFN (Thomas et al., 2018) and SE(3)-Transformer (Fuchs et al., 2020), the attention-based extension of TFN, are a family of SE(3)-equivariant model that performs the update for a representation $\hat{x}_i$ with a learned kernel $\phi$ utilizing spherical harmonics such that the model is equivariant with respect to $x_i$:

$$\hat{x}_i^{t+1} = \phi_u(\hat{x}_i^t) + \sum_{j \in \mathcal{V}, i \neq j} \phi_{x_j - x_i}(\hat{x}_j^t)$$

where $\phi_u$ is a simple projection matrix, and $\phi_{x_j - x_i}$ is a kernel matrix that learns to project $\hat{x}_j$ based on $(x_j - x_i)$. Since kernel $\phi_{x_j - x_i}$ consists of a learned radial function for the distance $d_{ij}$ and an angular preserve basis for $\frac{x_j - x_i}{d_{ij}}$, this formulation is in close relationship with Equation 3 as the angular preserve basis also captures the term $(x_i - x_j)$ while the learnable radial function is equivalent to the gradient estimator $f_v$. Therefore, such operation can also be interpreted as optimizing the same energy function of our two-atom model, but with higher-order equivariant representations involving special designed basis. Last but not least, since the optimized $x$ can only be obtained at the last step through a linear projection, the whole model is effectively performance only an one-step optimization.

# D  Data Preprocessing Details

## D.1  Reference Selection

There are three datasets we used in the experiment section: QM9, GEOM-QM9, and GEOM-Drugs.

For the molecular conformation *optimization* task, we choose to use QM9 instead of GEOM-QM9 because the conformers of it are computed with high fidelity DFT method. For the GEOM-Drugs dataset computed with a semi-empirical method, they include multiple low-energy stable conformers. If we compare the Boltzmann weights (proxy of energy stability) for the most likely conformers and the second likely conformers, we find many of them are pretty close, as shown in Figure S1a). In Figure S1b), we also showed an example where two conformers with very similar Boltzmann weight but drastically different conformation. Therefore, for GEOM-Drugs in the optimization setting, we use multiple reference conformers during the training and evaluation for GEOM-Drugs, but the model predicted conformers only need to match one of the references. In order to select similarly energetically preferred conformers, we select the set of target conformers as $\{X | P_X \geq 0.5 * P_{X^*}\}$ where $P_{X^*}$ is the Boltzmann weight for the most likely conformers.

For the molecular conformation *generation* task, we use both GEOM-QM9 and GEOM-Drugs as the dataset since they contain multiple references for each molecule. However, to filter out some of the rare conformers, we first sort reference conformers by their Boltzmann weights and set a threshold of the sum Boltzmann weight at 0.95 in the training set: $\{X_{1:n} | \sum_{i=1}^n P_{X_i} \geq 0.95, \sum_{i=1}^{n-1} P_{X_i} < 0.95, P_{X_1} \geq \cdots \geq P_{X_N}\}$, but still report metrics on all reference conformers as previous work does.

## D.2  Featurization

We construct the molecular graph $\mathcal{G}$ as a fully-connected graph to represent the molecules where the nodes are featured with atom type, aromatic, hybridization, and the number of hydrogens, while the edges are featured with the bond type (including no-bond), length of the shortest path in the graph, and whether the atom pair at both ends are in the same conjugated system. We perform an initial feature projection with $v$-update in each model to get the atom embedding and relationship embedding from these features. Similar to other works (Mansimov et al., 2019; Xu et al., 2020), we only model the heavy atoms for the molecular graph and conformer in this experiment.

# E  Experiment Details for Molecular Conformation Optimization

## E.1  Model Parametrization

**SE(3)-Transformer (Fuchs et al., 2020)**  is parametrized with the following hyper-parameters which maximize the GPU memory: `num_degrees=2`, `num_layers=9`, `hidden_dim=64`, and finally `n_heads=8`.

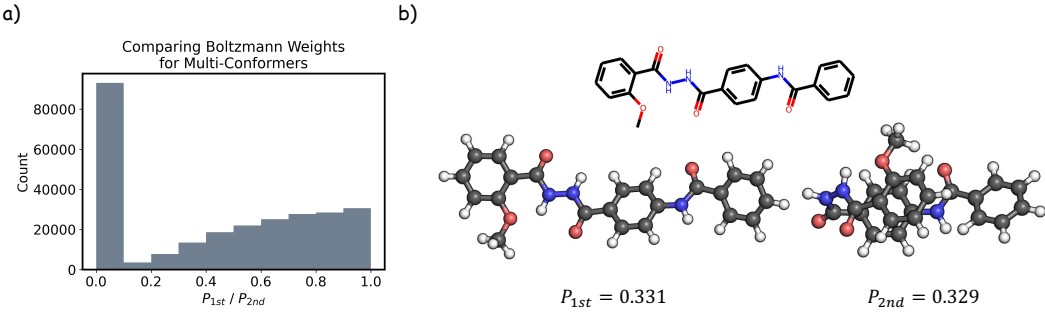

**Figure S1: Boltzmann Weights for Conformers in GEOM-Drugs Dataset.** For each molecule in the GEOM-Drugs dataset, multiple conformers are generated with a Boltzmann weight associated with them, representing their energy stability. **a)** We divide the Boltzmann weight of the most likely conformer with the one of the second likely conformer. The distribution of the ratio is shown, and many of the conformers have at least two top conformers that share a similar Boltzmann weight. **b)** We show an example where two conformers in very different shapes could share very similar Boltzmann weight.

**EGNN (Satorras et al., 2021)** is parametrized with the following hyper-parameters: `num_layers=9`, `hidden_dim=256`, and the distance square features are expanded using an RBF kernel with 50 basis between 0Å - 10Å.

**Ours -TwoAtom and -Ext$_v$** contain three optimization blocks sharing the parameters, and within each optimization block, there are three layers of $x$ and $v$ updates. The neural network in these updates are featured as a Transformer (Vaswani et al., 2017) with `hidden_dim=256` and `n_heads=16`. The key/value embedding and attention score are generated through a 2-layer MLP with LayerNorm and ReLU. In addition, an additional interaction weight is introduced to model different energy contribution of the atom pairs which is simply modeled as $\alpha_{ij} = \text{sigmoid}(\text{MLP}(d_{ij}))$. Last but not least, we implement a dynamic cutoff where we ignore the energy contribution between atom pairs whose euclidean distance is less than 10Å. Similar to the EGNN implementation above, all distance features are also expanded with the same RBF kernel. Figure S2 shows the overall architecture.

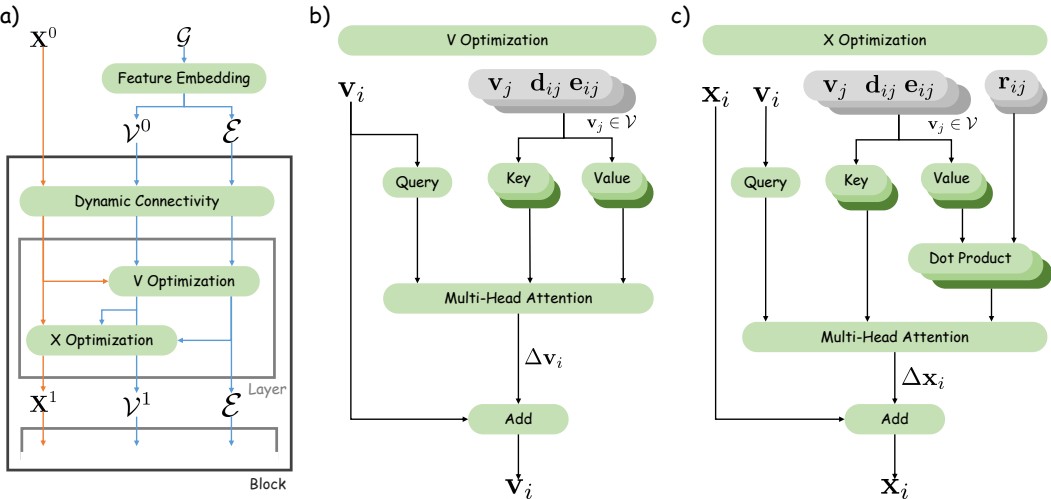

**Figure S2: Model Architecture for Ours-TwoAtom.** **a)** Overall architecture with one optimization block consisting multiple optimization layer. **b)** Optimization for $v$ where $d_{ij} = \|x_i - x_j\|$. **c)** Optimization for $x$ where $\mathbf{r}_{ij} = x_i - x_j$.

**Ours-ThreeAtom** is a three-atom model extension of Ours-Ext$_v$, and therefore follow the same model hyper-parameters as above. Since we consider the bond angles $\angle_{jik}$ and $\angle_{kij}$ the same, the neural networks $f_x$, $g_x$ and $f_v$ need to satisfy permutation invariant between $ij$ and $ik$. Therefore,

for these three functions, we construct them as the following:

$$f = [f(d_{ij}, d_{ik}, \langle \boldsymbol{r}_{ij}, \boldsymbol{r}_{ik} \rangle, \mathcal{V}_{ijk}, \mathcal{E}_{ijk}) + f(d_{ik}, d_{ij}, \langle \boldsymbol{r}_{ik}, \boldsymbol{r}_{ij} \rangle, \mathcal{V}_{ikj}, \mathcal{E}_{ikj})]/2$$

In addition, we generalize the three-atom model to the two-atom case by including the triplet $(i, j, k)$ where $j = k$. While treating the molecule as a fully connected graph allows us to model the non-local relationship, the challenge is that the number of atom triplets grows very quickly with the size of the molecule. Since the bond angle in a conjugated ring system with no rotatable bond is not super interesting, we trim down the number of atom triplets by only considering $(i, j, k)$ triplet where $e_{ij}$ or $e_{ik}$ represents a rotatable bond.

## E.2 Training Procedures

Our proposed models and other machine learning baselines are all trained in the same procedure.

We sample a batch size of 128 molecules for the QM9 dataset and a batch size of 32 molecules for the GEOM-Drugs dataset during the training. In one training batch, one initial conformer is randomly sampled for each molecule from a pool of ten pre-initialized conformers from RDKit+MMFF. Since we expect the model to stop the optimization when the input conformer is already stable, we also sample reference conformer as the initial input conformer with a probability $p = 0.05$.

The model is trained end-to-end by the L2 loss between the distance matrices for the predicted conformer and the reference conformer via gradient descent method Adam Kingma & Ba (2015). The hyper-parameter settings for the optimizer are `init_learning_rate=0.001`, `betas=(0.95, 0.999)`, and `clip_gradient_norm=8`. We also schedule to decay the learning rate exponentially with a factor of 0.5 and a minimum learning rate of 0.00001. The learning rate is decayed if there is no improvement for the validation loss in 8 consecutive evaluations, and the training will be completely stopped if no improvement is found for 20 consecutive evaluations. The evaluation is done for every 2000 training steps/batches. We tried a small number of learning rate schedules based on the validation set to make sure the training is sufficient. Gradient norm is clipped to make the training stable. Other hyper-parameters are set following the common choices.

## E.3 Computation Resource and Training Time

We train all conformer optimization models for the QM9 dataset with one NVIDIA GeForce GTX 1080 GPU and use one GeForce GTX 3090 GPU for the GEOM-Drugs dataset. All the models can be trained within 36-48 hours except the Ours-ThreeAtom model, which takes a longer time.

## E.4 Code and Data Availability

We train all conformer optimization models for the QM9 dataset with one NVIDIA GeForce GTX 1080 GPU and one GeForce GTX 3090 GPU for the GEOM-Drugs dataset. All the models can be trained within 36-48 hours except the Ours-ThreeAtom model, which takes longer.

# F Experiment Details for Downstream Applications

## F.1 Correlation for the HOMO-LUMO Gap

For this study, we sample 400 molecules from the test dataset of the conformer optimization experiment (Sec. 4.1) for the quantum mechanic calculation. Since the dataset and the model is heavy atom only, we infer the coordinate of hydrogens via RDKit (Riniker & Landrum, 2015). The HOMO and LUMO energy for the reference and two-atom model predicted conformers are finally calculated via Psi4 (Smith et al., 2020) with the MP2/6-311++G(d,p) basis set.

## F.2 Training Neural Networks with Predicted Conformers

In this study, we train the standard SchNet (Schütt et al., 2017) on the common QM9 property prediction benchmark with the data and splits provided by Anderson et al. (2019). However, we want to train and evaluate the model with the generated conformers instead of using the reference conformers. Since SchNet (Schütt et al., 2017) requires the hydrogen atom for its full potential and the benchmark dataset follows a different split as our conformer optimization task, we train a new

two-atom model on the training split of this benchmark with the hydrogen atom included and use this new model to generate conformers.

In our study, SchNet contains 7 layers and a hidden dimension of 128. To expand the distance feature, we use 50 Gaussian basis, and we construct the molecular graph with a distance cutoff of 10 Å. We use L1 Loss to train the model end to end with Adam (Kingma & Ba, 2015) (`weight_decay=5e-16`, `batch_size=96`, and `num_epochs=1000`). The learning rate starts between `5e-4` and `1e-6` for different tasks and follows a cosine decay.

Since the single reference conformer does not constrain us, we consider multiple conformers in this study. RDKit+MMFF first estimates multiple initial conformers, and different predicted conformers are generated from these initial conformers with our two-atom model and the baseline EGNN model. To select for these initial conformers, we first sample 10 conformers with RDKit+MMFF, and the top 5 conformers with the lowest energy are selected. One of the five conformers is sampled randomly at each batch during training, but the average prediction from the five conformers is reported as the model estimate during inference.

## G  EXPERIMENT DETAILS FOR MOLECULAR CONFORMATION GENERATION

We follow the setting in Shi et al. (2021) and report baseline results from their paper. The metrics we used are defined as follows:

$$\text{COV}(S_g, S_r) = \frac{1}{|S_r|} |\{\boldsymbol{X}^* \in S_r | \text{RMSD}(\boldsymbol{X}, \boldsymbol{X}^*) \leq \delta, \boldsymbol{X} \in S_g\}|$$

$$\text{MIS}(S_g, S_r) = \frac{1}{|S_g|} |\{\boldsymbol{X} \in S_g | \text{RMSD}(\boldsymbol{X}, \boldsymbol{X}^*) > \delta, \forall \boldsymbol{X}^* \in S_r\}|$$

$$\text{MAT}(S_g, S_r) = \frac{1}{|S_r|} \sum_{\boldsymbol{X}^* \in S_r} \min_{\boldsymbol{X} \in S_g} \text{RMSD}(\boldsymbol{X}, \boldsymbol{X}^*)$$

where $\text{RMSD}(\boldsymbol{X}, \boldsymbol{X}^*) = \min_{\Phi}(\frac{1}{n} \sum_{i=1}^{n} \|\Phi(\boldsymbol{X}_i) - \boldsymbol{X}_i^*\|^2)^{\frac{1}{2}}$. $n$ is the number of heavy atoms and $\Phi$ is an alignment function which aligns two conformers with rotational and translational operations. The thresholds of COV and MIS are $\delta = 0.5$ and $\delta = 1.25$ for GEOM-QM9 and the larger GEOM-Drugs respectively, which are same to Shi et al. (2021).

We used the same setup as the two-atom model in the conformation optimization setting with the same hyper-parameters for this experiment. Tor train the model, we use the optimal transport loss during the training as described in Eq. 7 and choose $K = 5$. In Table. 3, the conformers in both training and evaluation phases of *Ours-Random* are initialized with random noise, which is drawn from a Gaussian $\mathcal{N}(0, \sigma^2)$ and $\sigma = 0.028 * |\mathcal{V}|$. On the other hand, *Ours-RDKit* are initialized with permuted reference conformers during training, where the permutation is drawn from a Gaussian $\mathcal{N}(0, \sigma)$ and $\sigma = 0.5$ for GEOM-QM9 and $\sigma = 1.0$ for GEOM-Drugs to account for the average volume difference. In evaluation, we apply the model with initialization from RDKit since ground truth reference is not available.

## H  ADDITIONAL EXPERIMENTS

### H.1  CONFORMER OPTIMIZATION WITH RANDOM INITIALIZATION

While our *optimization* experiments are done with an initial conformer estimated by RDKit+MMFF, the initial conformer can be generated elsewhere, and even from other generative models like CGCF Xu et al. (2020). In the simplest case, we could also initiate the conformers from random points for the model training and evaluation where each coordinate value is drawn from a Gaussian $\mathcal{N}(0, \sigma^2)$ and $\sigma = 0.1 * |\mathcal{V}|$. As shown in Table S1, while the model performance is less satisfactory with the randomly initialized conformers, the predicted conformers are still quite accurate and even more accurate than the ones optimized by the hand-designed force-field method. This also shows the potential of our methods as the model prediction can be even more accurate if better initialization is possible.

| Dataset Model | QM9 | | GEOM-Drugs | |
|---|---|---|---|---|
| | mean RMSD ($\downarrow$) | median RMSD ($\downarrow$) | mean RMSD ($\downarrow$) | median RMSD ($\downarrow$) |
| RDKit+MMFF | $0.3872 \pm 0.0029$ | $0.2756 \pm 0.0075$ | $1.7913 \pm 0.0030$ | $1.6433 \pm 0.0097$ |
| Ours-TwoAtom | $0.1415 \pm 0.0004$ | $0.0534 \pm 0.0002$ | $0.8839 \pm 0.0014$ | $0.7733 \pm 0.0026$ |
| Ours-TwoAtom-Random | $0.2256 \pm 0.0008$ | $0.1602 \pm 0.0020$ | $1.3026 \pm 0.0037$ | $1.2087 \pm 0.0049$ |

**Table S1:** RMSD when training and evaluating our optimization models with random initialization instead of RDKit+MMFF. The confidence interval is calculated from the inference with ten different initializations.

## H.2 PREDICTING ENSEMBLE PROPERTY

Here we perform the same ensemble property prediction as Table 3 of Shi et al. (2021), where they measure how well the generated conformers can capture the population statistics of electronic property of the reference set. The authors are very kind and provide us with the code and the 30 molecules used in their experiments. We reproduce the results here for future references.

| Method | $\overline{E}$ | $E_{\min}$ | $\overline{\Delta\epsilon}$ | $\Delta\epsilon_{\min}$ | $\Delta\epsilon_{\max}$ |
|---|---|---|---|---|---|
| RDKit | $1.03\pm0.34$ | $0.67\pm0.21$ | $0.38\pm0.22$ | $0.36\pm0.47$ | $1.08\pm0.98$ |
| ConfGF | $1.67\pm3.48$ | $0.16\pm0.13$ | $0.47\pm0.44$ | $0.13\pm0.09$ | $2.29\pm2.59$ |
| Ours-Random | $2.63\pm6.25$ | $1.71\pm3.03$ | $0.62\pm0.65$ | $0.37\pm0.24$ | $1.03\pm1.21$ |

**Table S2:** The mean and standard deviation of the absolute errors of predicted ensemble properties. Unit: eV.

## H.3 REPRESENTATION LEARNING

While we have been focusing on optimizing molecular conformation, we can also apply our SE(3)-equivariant model for molecular representation learning, even when the native conformer is already given. In this experiment, we treat the conformer $X$ as a constant and only update the internal atom representation $\mathcal{V}$ under the framework of implicit energy minimization. Following the same setup as Satorras et al. (2021), we compare our model against other models on the QM9 property prediction benchmark. As shown in Table S3, our $v$-update-only model can achieve a comparable performance with state-of-the-art models as well. The consistently good performance again demonstrates the generalizability of our framework where powerful SE(3)-equivariant *and* -invariant models can be derived with different definitions of the energy function.

| Task Model | $\alpha$ bohr$^3$ | $\Delta\epsilon$ meV | $\epsilon_{\text{HOMO}}$ meV | $\epsilon_{\text{LUMO}}$ meV | $\mu$ D | $C_\nu$ cal/mol K |
|---|---|---|---|---|---|---|
| WaveScatt (Hirn et al., 2017) | 0.160 | 118 | 85 | 76 | 0.340 | 0.049 |
| NMP (Gilmer et al., 2017) | 0.092 | 69 | 43 | 38 | **0.030** | 0.040 |
| SchNet (Schütt et al., 2017) | 0.235 | 63 | 41 | 34 | 0.033 | 0.033 |
| Cormorant (Anderson et al., 2019) | 0.085 | 61 | 34 | 38 | 0.038 | **0.026** |
| L1Net (Miller et al., 2020) | 0.088 | 68 | 46 | 35 | 0.043 | **0.031** |
| LieConv (Finzi et al., 2020) | **0.084** | **49** | **30** | **25** | 0.032 | 0.038 |
| TFN (Thomas et al., 2018) | 0.223 | 58 | 40 | 38 | 0.064 | 0.101 |
| SE(3)-Tr. (Fuchs et al., 2020) | 0.142 | **53** | 35 | 33 | 0.051 | 0.054 |
| EGNN (Satorras et al., 2021) | **0.071** | **48** | **29** | **25** | **0.029** | **0.031** |
| Ours | **0.078** | **53** | **31** | **27** | **0.019** | 0.037 |

**Table S3:** MAE for QM9 molecular property prediction benchmark comparing to other non-equivariant (top) and equivirant (bottom) models. The top-3 performances for each task are **bolded**.

## I    Study of Model Ablation and Variation

### I.1    Effects of graph connectivity

While our model is based on the attention mechanism on the whole graph, we can also define the graph connectivity to reduce the computational complexity. Performing attention directly on the molecular graph may be inefficient. One possible solution is to perform attention on the augmented graph as (Xu et al., 2020; Simm & Hernandez-Lobato, 2020) where the molecular graph is augmented with *virtual* bonds connecting atoms that are 2 or 3 hops away in the original graph. However, such node distances are sometimes not informative enough since two atoms far apart in graph distance may be close in the 3D space. An alternative way is to construct a fully connected graph to capture all long-range dependencies, but it could introduce noise to the attention process. Our model dynamically calculates the graph connectivity based on the euclidean distances between atom pairs/triplets at each intermediate optimization step. Specifically, we exclude the contributions for pairs/triplets if they are too far from each other in the 3D space and only consider interaction inside this radius cutoff.

Table S4 shows that dynamic connectivity is consistently better than other approaches for both the QM9 and GEOM-Drugs datasets. Although fully connected attention works better than augmented attention on the QM9 dataset, it is worse than the augmented graph on GEOM-Drugs, even with more computational complexity. The performance degradation here could be caused by the noise from unrelated atom pairs/triplets. In contrast, our dynamic connectivity can take advantage of the predicted conformers and construct more informative connectivity than the augmented approach based on the graph distance.

| Dataset Model | QM9 | | GEOM-Drugs | |
|---|---|---|---|---|
| | mean RMSD | median RMSD | mean RMSD | median RMSD |
| Augmented | 0.1476 | 0.0568 | 0.9194 | 0.8136 |
| Fully-Connected | 0.1456 | 0.0569 | 0.9244 | 0.8140 |
| Dynamically-Connected | 0.1415 | 0.0534 | 0.8839 | 0.7733 |

**Table S4:** Study of Model Ablation and Variation for Molecular Graph Connectivity.

### I.2    Effects of the parameterization for interaction weight

While the dynamic connection discussed in the above section treats the interaction between atoms in a binary fashion, we also consider their interaction weight fractionally. In EGNNSatorras et al. (2021), it estimates the interaction weights with a linear layer which takes $v$ as inputs. In our model, we propose a prediction layer for weights of interaction, which considers the distance between an atom pair and their atom type.

In Table S5, we test different approaches to parameterize the interaction weights. While we find the Per-layer $v$ (EGNN-like) performs worse than others, the effects of distance $d$ based prediction net vary from dataset to dataset. In the models reported in the main text, we use Per-block $d$ for all variants, but a Per-layer $d$ setup could improve the performance of the GEOM-Drugs dataset. Thus, we could tune this design from dataset to dataset. We leave further exploration to future work.

| Dataset Model | QM9 | | GEOM-Drugs | |
|---|---|---|---|---|
| | mean RMSD | median RMSD | mean RMSD | median RMSD |
| None | 0.1456 | 0.0582 | 0.8780 | 0.7618 |
| Per-layer $v$ | 0.1590 | 0.0693 | 0.8918 | 0.7719 |
| Per-layer $d$ | 0.1427 | 0.0555 | 0.8529 | 0.7287 |
| Per-block $d$ | 0.1415 | 0.0534 | 0.8839 | 0.7733 |

**Table S5:** Study of Model Ablation and Variation for Interaction Weight Parameterization. None refers to no interaction prediction net; Per-layer $v$ is an EGNN-like prediction net; Per-layer $d$ and Per-block $d$ directly take intermediate atomic distances as inputs and perform prediction at each layer or each block separately.

## I.3 EFFECTS OF PARAMETER SHARING

For our models reported in the main text, we implement three optimization blocks that share their parameters while each block contains three layers of $x$ and $v$ updates that do not share their parameters. However, one could push this to either extreme in terms of parameter sharing. For the same number of total update layers, we could have either nine shared optimization blocks with only one layer per block or a single optimization block with a total of nine layers that do not share parameters at all. It is worth mentioning that all these parameter sharing designs make sense since they correspond to the different hypotheses of the shape of the gradient field. For example, if the gradient field is relatively smooth, sharing parameters across all blocks could also perform well. We report the performance for these two extreme cases in Table S6 and found our hybrid design strikes the right balance between the two.

| Dataset Model | QM9 | | GEOM-Drugs | |
|---|---|---|---|---|
| | mean RMSD | median RMSD | mean RMSD | median RMSD |
| 9-Block × 1-Layer | 0.1505 | 0.0597 | 0.8874 | 0.7727 |
| 1-Block × 9-Layer | 0.1395 | 0.0527 | 0.9750 | 0.8792 |
| 3-Block × 3-Layer | 0.1415 | 0.0534 | 0.8839 | 0.7733 |

**Table S6:** Study of Model Variation for Parameter Sharing.

## I.4 EFFECTS OF ITERATIVE UPDATE

For our models in the main text, the coordinate $x$ and the internal representation $v$ are updated in an iterative fashion, where the update for $x^{t+1}$ depends on $v^{t+1}$. However, we can also update them in a parallel fashion as EGNN does, where the update for $x^{t+1}$ will depend on $v^t$ instead. These are both reasonable approaches depending on the optimization strategies.

Empirical results in Table S7 show that optimizing $x$ and $v$ iteratively is slightly better than optimizing them in a synchronized fashion.

| Dataset Model | QM9 | | GEOM-Drugs | |
|---|---|---|---|---|
| | mean RMSD | median RMSD | mean RMSD | median RMSD |
| Parallel Update | 0.1483 | 0.0584 | 0.9588 | 0.8515 |
| Iterative Update | 0.1415 | 0.0534 | 0.8839 | 0.7733 |

**Table S7:** Study of Model Variation of Iterative and Parallel Update.

## I.5 COMPARISON WITH EGNN

| Dataset Model | Designs | | | | QM9 | | GEOM-Drugs | |
|---|---|---|---|---|---|---|---|---|
| | Tfm-BB | IW | PS | IU | mean RMSD | median RMSD | mean RMSD | median RMSD |
| EGNN | | | | | 0.2101 | 0.1356 | 1.0405 | 0.9598 |
| EGNN-Tfm | ✓ | | | | 0.2049 | 0.1209 | 1.0334 | 0.9458 |
| Per-layer IW | ✓ | | ✓ | ✓ | 0.1590 | 0.0693 | 0.8918 | 0.7719 |
| Non-Share Para. | ✓ | ✓ | | ✓ | 0.1395 | 0.0527 | 0.9750 | 0.8792 |
| Parallel Update | ✓ | ✓ | ✓ | | 0.1483 | 0.0584 | 0.9588 | 0.8515 |
| Ours(Full) | ✓ | ✓ | ✓ | ✓ | 0.1415 | 0.0534 | 0.8839 | 0.7733 |

**Table S8:** Ablation Study Comparing to EGNN. *Tfm-BB* denotes the transformer backbone. *IW* denotes interaction weight, which is marked if per-block interaction weight is adopted, otherwise per-layer interaction weight is adopted. *PS* denotes parameter sharing. *IU* denotes parallel update.

Finally, we summarize our different designs and their effects comparing to EGNN in Table S8. There are a couple of differences between our methods and EGNN from the unrolled optimization

formulation including per-block interaction weight (I.2), shared parameters (I.3), and iterative updates (I.4). These differences make up some of the improvements together. On the parameterization side, arguably the biggest differences are that EGNN uses an MLP architecture while ours uses a Transformer-like architecture. However, switching their MLP architecture with a Transformer-like architecture provides a little bit of lift, but it's not the major contributor.

## J   ABLATION STUDIES IN CONFORMATION GENERATION

| Dataset | GEOM-Drugs | | | | | |
|---|---|---|---|---|---|---|
| Metric | COV(%) ($\uparrow$) | | MIS(%) ($\downarrow$) | | MAT(Å) ($\downarrow$) | |
| | Mean | Median | Mean | Median | Mean | Median |
| Multi-step Eval | 72.96 | 76.73 | 37.78 | 28.88 | 1.1031 | 1.0853 |
| EGNN-bb | 77.09 | 82.57 | 23.88 | 11.58 | 1.0491 | 1.0514 |
| Non OT-Loss | 59.67 | 60.96 | 16.41 | 1.82 | 1.2443 | 1.2223 |
| Full model | 76.50 | 83.78 | 31.40 | 23.03 | 1.0694 | 1.0583 |

**Table S9:** Ablation Study in Conformation Generation. *Multi-step Eval* means the model is trained to learn the part optimization process and evaluated with multiple steps to achieve the entire optimization. *EGNN-bb* denotes the model uses EGNN as the backbone. *Non OT-Loss* means the model is trained to minimize the L2 loss between the generated conformers and their closest reference conformers, instead of using the optimal transport loss.

We introduce ablation studies to further study the importance of different components of our proposed method in the generative setting. We find the formulation of framing the generative task as a fixed-step unroll optimization process using an SE(3)-equivariance model is the more important bit, as we only see a very small performance regression when switching our backbone optimization model to EGNN. On the other hand, ConfGF is proposed to model one-step optimization and relies on Langevin dynamics for performing the entire sequence of optimization. Additionally, the optimal transport loss also helps quite a bit, as it encourages the model to optimize the conformers towards different directions instead of collapsing the mode.

## K   INFLUENCE OF NUMBER OF UNROLLED OPTIMIZATION STEPS

We test the optimization performance on the QM9 dataset with different numbers of optimization steps. We observe a diminishing return for steps >4 since the "gradient" would be small if we are close to the solution. Therefore, we believe the method is not sensitive to the number of steps as long as a sufficient number of steps are taken. In all our experiments, we perform a fixed number of 9 optimization steps.

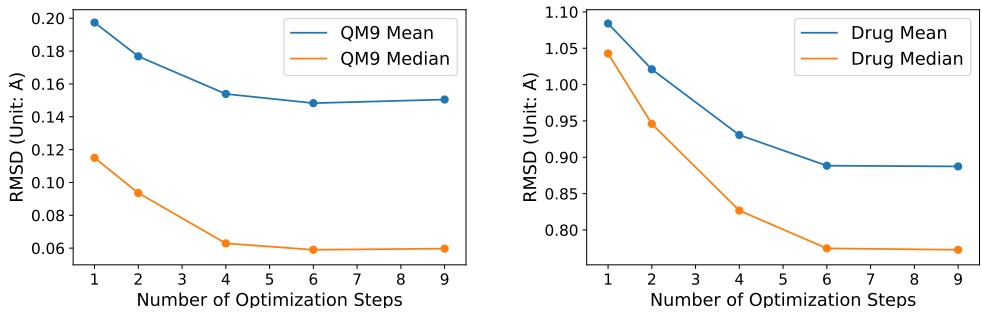

**Figure S3:** Influence of Number of Unrolled Optimization Steps

## L   MORE EXAMPLES OF MODEL-OPTIMIZED CONFORMERS

Please see next page.

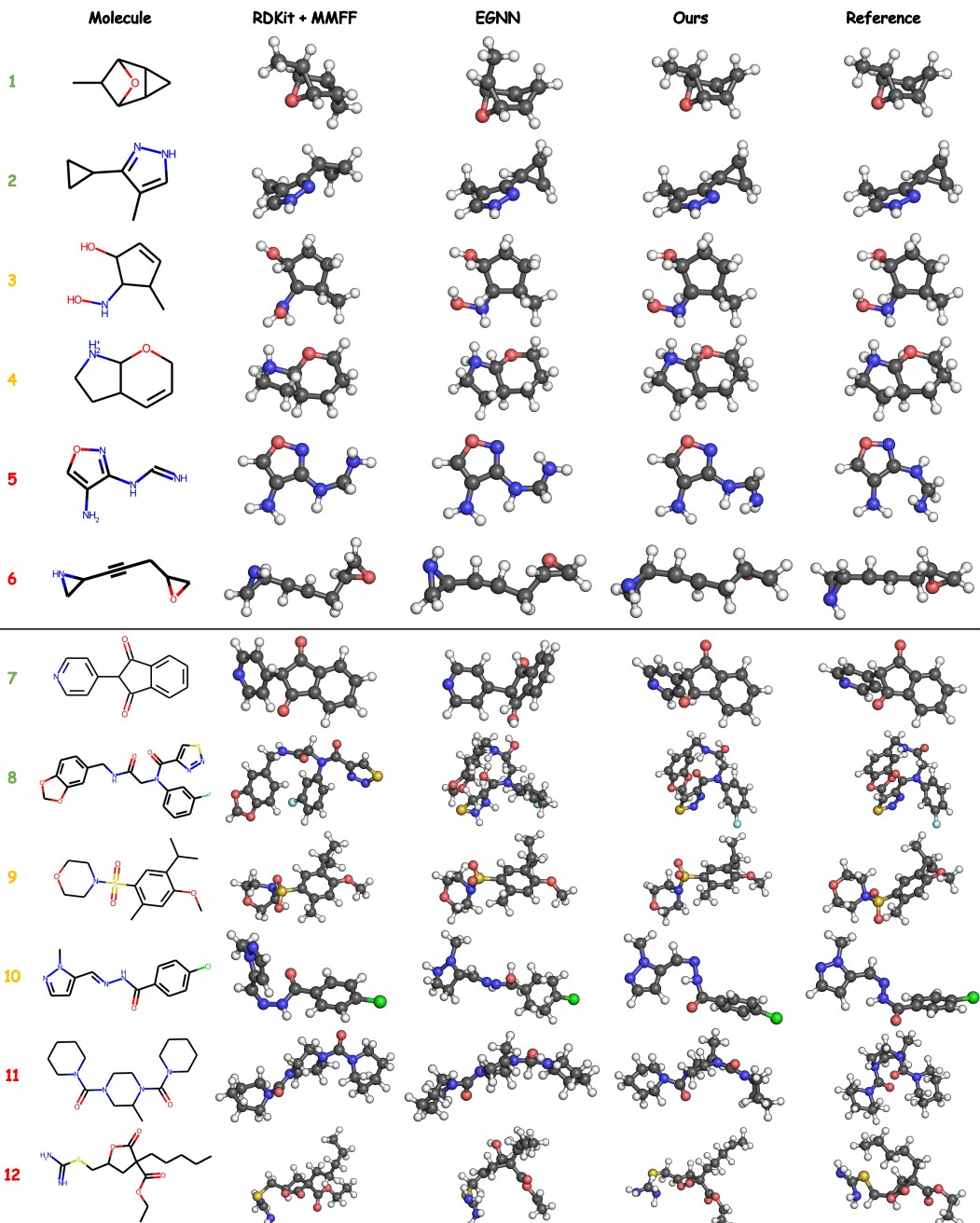

**Figure S4: More Examples of Model-Predicted Conformers.** Molecule 1-6 are sampled from the QM9 dataset, and the larger molecule 7-12 are sampled from the GEOM-Drugs dataset. The reference conformer, as well as the initial conformations from RDKit+MMFF, are also shown. Green IDs are the successful cases where the predicted conformers' RMSD are $\leq 10\%$ of the population. Red IDs are the failure cases where the predicted conformers' RMSD are $\geq 90\%$ of the population. Yellow IDs are the ones with RMSD between 45% and 55% of the population.

