# OpenReview forum: "Energy-Inspired Molecular Conformation Optimization"
_ICLR.cc/2022/Conference — ICLR 2022 Poster_

### Official Review · Reviewer_jLPL · 2021-10-25

**Correctness:** 4
**Technical Novelty And Significance:** 3
**Empirical Novelty And Significance:** 2
**Recommendation:** 8
**Confidence:** 3

**Main Review:**

# 1. Positives
i. The paper has strong empirical results.
By considering both optimization and generation baselines, the authors are able to compare against a wide range of previous baselines over several different forms of evaluation. They obtain improved performance over these models using multiple different metrics. This improved performance also seems to be obtained with reduced compute costs (e.g., approx. 170x speedup over ConfGF).

ii. I found the paper easy to follow.
Figure 1 is helpful in understanding the approach. I found Section 3 interesting and well-organized; describing previous methods through considering the implied form of the potential energy function (see also Appendix C) seems like a useful lens in which to compare and view different models.


# 2. Negatives
i. It would be nice to have been provided with some evidence for what aspect of the proposed approach is important for improving upon the previous SOTA ML model, ConfGF, in Table 3, given that (at a high-level) ConfGF is also predicting the gradient of the energy function. For instance, is it because of the unrolled optimization routine, different architectures, the optimal transport loss, etc.? Could this be teased out with an ablation study?

ii. The number of time steps (i.e. the max value of $t$) used/required in the different experiments is not very well discussed/evaluated. Questions I would like to see answered include: Is the authors' model sensitive to this value? Does the value differ for the two datasets? How can one choose this value?


# 3. Clarification Questions to the Authors

i. Table S1 indicates that the RDKIT initialization is important for performance. I just wanted to check that you also used RDKIT initializations for the baseline approaches you compared to (where applicable)? As opposed to RDKit initializations, have you thought about predicting the initial positions?

ii. I'm surprised that in Table 1 "Ours-TwoAtom" does so much better than EGNN, given on p.5 you describe the similarities between these models. Do you have any intuition for why this is?

iii. Sorry if I missed this, but in Table 3 are you using the two atom or three atom model?

iv. In Table 2 is SchNet really the best reference model? From Table S2 it seems that one can do much better with an equivariant model (when you know the conformation).

v. How do you choose the step size in Eqn 1, $\gamma$?





# 4. Very minor comments (e.g., typos)
i. The grammar/typos could be fixed in a few places, e.g.:
- p.1 "could be required to encode the redundancy observed in such matrix" --> "could be required to encode the redundancy observed in such _matrices_" ?
- Fig1, caption: "the gradient updates for the 3D coordinate long with the updated atom" --> "the gradient updates for the 3D coordinate _along_ with the updated atom"
- p.5 "perspective of neural energy minimization, these are equivalent to optimizing an energy function that depends on both the atom presentations and inter-atomic distances just like our basic" --> "perspective of neural energy minimization, these are equivalent to optimizing an energy function that depends on both the atom _representations_ and inter-atomic distances just like our basic" ?
- p.15 "we ﬁnd many of them are pretty closed, as" --> "we ﬁnd many of them are pretty _close_, as"

ii. On p.9 when introducing terms COV, MIS, and MAT you could refer to their definitions in Appendix G. Also what value does $\delta$ take? (apologize if I missed this)

iii. The experiments in Appendix I are interesting and impressive! Maybe it would have been nice to have more explicitly referred to this section in the main paper?

iv. How were the hyperparameters described in Appendix E chosen?



**Summary Of The Paper:**

This paper proposes a new method to generate molecular conformations (i.e., the spatial arrangements of atoms belonging to a given molecule). The method works by updating initial atom positions (which can be initialized either randomly or using an alternative conformer generation method) through an iterative process. Each iteration involves predicting the gradients associated with atoms' positions and then changing their coordinates by taking a step in this direction (Eqn. 1 & Fig. 1b).

The authors:
1. show how the parameterization of the gradients relates to how one models the conformational energy (Section 3.3);
2. demonstrate competitive performance on optimization (finding a single best conformer; Section 4.1) and generation (finding a set of relatively stable conformers; Section 4.2) tasks compared to previous benchmarks. This is judged in terms of (a) how well the generated conformers match up to ground truth conformers (e.g., Table 1) as well as (b) how well the generated conformers enable the prediction of molecular properties (Table 2).


**Summary Of The Review:**

Overall, I thought this was an interesting paper and the proposed method achieves impressive empirical results. However, for now, I have gone with an overall score of 6, as I would like to see some of the experimental aspects better resolved (see 2.i-ii & 3.i-ii). Happy to consider raising my score if this gets addressed in the rebuttal!

I went with a lower confidence score as I have not gone through the proofs in the appendix in detail and only have a higher-level knowledge of some of the pieces of related work.

##  Rationale behind the significance scores
I put _"3: The contributions are significant and somewhat new. Aspects of the contributions exist in prior work."_ for the technical contributions question below. I did not choose the top score, given that Shi et al. (2021) also consider parameterizing the gradients of the energy function directly. (I realize that there are differences however, such as the training routine and optimal transport loss).

I put _"2: The contributions are only marginally significant or novel."_ for the empirical novelty and significance. This paper's method is evaluated on existing datasets (Ramakrishnan et al., 2014; Axelrod & Gomez-Bombarelli, 2020) in a way that is similar to existing work. To introduce a new empirical evaluation the authors could consider evaluating how well the models trained on one dataset transfer to another or consider directly comparing the predicted gradients (from the NN) to gradients derived from traditional approaches. (Although in my opinion making more of a technical contribution and less of an empirical one (or vice versa) is fine).

---

> ### Author Response · Authors · 2021-11-16
> **Author Response to Reviewer jLPL**
>
> We thank the reviewer for the feedback and suggestions. The transparency on the rationale behind the significance scores is much appreciated as well :) Please see below for our responses to the comments.
>
> **Negatives**
>
> **Q1 - “It would be nice to have been provided with some evidence for what aspect of the proposed approach is important for improving upon the previous SOTA ML model, ConfGF, in Table 3, given that (at a high-level) ConfGF is also predicting the gradient of the energy function.”**
>
> A1 - Thank you for the suggestions, and we agree! In Appendix J, we introduce new ablation studies to further study the importance of different components of our proposed method in the generative setting (also attached below).
>
> |  GEOM-Drugs               | COV Mean | COV Median | MIS Mean | MIS Median | MAT Mean | MAT Median |
> |-----------------|----------|------------|----------|------------|----------|------------|
> | Multi-step Eval | 72.96    | 76.73      | 37.78    | 28.88      | 1.1031   | 1.0853     |
> | EGNN-bb         | 77.09    | 82.57      | 23.88    | 11.58      | 1.0491   | 1.0514     |
> | Non OT-Loss     | 59.67    | 60.96      | 16.41    | 1.82       | 1.2443   | 1.2253     |
> | Full model      | 76.50    | 83.78      | 31.40    | 23.03      | 1.0694   | 1.0583     |
>
> We have the following findings:
> - The formulation of framing the generative task as a fixed-step unroll optimization process using an SE(3)-equivariance model is the more important bit as we only see a very small performance regression when switching our backbone optimization model to EGNN. On the other hand, ConfGF is proposed to model one-step optimization and relies on Langevin dynamics for performing the entire sequence of optimization.
> - The optimal transport loss also helps quite a bit, as it encourages the model to optimize the conformers towards different directions instead of collapsing the mode.
>
> **Q2 - “The number of time steps (i.e. the max value of ) used/required in the different experiments is not very well discussed/evaluated. Questions I would like to see answered include: Is the authors' model sensitive to this value? Does the value differ for the two datasets? How can one choose this value?”**
>
> A2 - This is a great question that we should have studied! In the updated Appendix K, we added a new experiment where we test the optimization performance on the QM9 dataset with different numbers of optimization steps. We observe a diminishing return for steps > 4 since the “gradient” would be small if we are close to the solution. Therefore, we believe the method is not sensitive to the number of steps as long as a sufficient number of steps are taken. In all our experiments, we perform a fixed number of 9 optimization steps. In practice, one might want to choose a large enough step size that is suitable to their hardware limits. Again, thank you for this great suggestion, and we have also updated the main text to highlight the finding.
>
> **Clarifications**
>
> **Q3 - “I just wanted to check that you also used RDKit initializations for the baseline approaches you compared to (where applicable)? As opposed to RDKit initializations, have you thought about predicting the initial positions?”**
>
> A3 - Yes, you are correct. All the baselines use the same RDKit initialization in the optimization experiments. We indeed have thought of initializing the conformers with predicted conformers. In the generative setting, a simple test where initial conformers are generated with ConfGF instead of random shows that our model can benefit from better initialization. This indicates the possibility of marrying other out-the-shelf conformation prediction models with our optimization formulation to achieve better performance.
>
> |  GEOM-Drugs               | COV Mean | COV Median | MAT Mean | MAT Median |
> |-----------------|----------|------------|----------|------------|
> | Ours w/ Random init     | 76.50    | 83.78      | 1.0694   | 1.0583     |
> | Ours w/ ConfGF init    | 77.54 | 84.03 | 1.0613 | 1.0533 |
>
> **Q4 - “I'm surprised that in Table 1 "Ours-TwoAtom" does so much better than EGNN, given on p.5 you describe the similarities between these models. Do you have any intuition for why this is?”**
>
> A4 - We summarize the different designs and their effects comparing to EGNN in Appendix I.5. Besides the parametrization, there are a couple of differences between our methods and EGNN coming from the neural energy minimization formulation including per-block interaction weight (Appendix I.2), shared parameters (Appendix I.3), and iterative updates (Appendix I.4). These differences make up some of the improvements together. On the parameterization side, arguably the biggest differences are that EGNN uses an MLP architecture while ours uses a Transformer-like architecture. However, switching their MLP architecture with a Transformer-like architecture provides a little bit of lift, but it’s not the major contributor.

---

> > ### Author Response · Authors · 2021-11-16
> > **Author Response to Reviewer jLPL - Cont.**
> >
> > **Q5 - “in Table 3 are you using the two-atom or three-atom model?”**
> >
> > A5 - We use the two-atom model here for simplicity, as pointed out at the beginning of Section 4, the purpose of the experiment/benchmark in the generative setting (Section 4.2) is to show that our proposed optimization formulation can be easily extended from the optimization setting to the generative setting, and achieve comparable performances with the state-of-the-art. Since the claim here is more about the versatility of the optimization formulation, we just use the simpler two-atom model here and didn’t even change the hyper-parameter following the same values from the optimization experiment.
> >
> > **Q6 - “In Table 2 is SchNet really the best reference model? From Table S2 it seems that one can do much better with an equivariant model (when you know the conformation).”**
> >
> > A6 - We choose SchNet here because it is one of the most battle-tested and common backbone models the field uses. In addition, SchNet also takes into account the conformation information during the modeling, so the quality of the conformation would affect the quality of the inductive biases.
> >
> > **Q7 - "How do you choose the step size in Eqn 1?”**
> >
> > A7 - Interesting question! We did not explicitly encode the step size, so we can think about it as either 1) the step size is simply 1, or 2) the step size is estimated by the model along with the gradient. In the early phase of exploratory research, we do try to explicitly model the step size with a global trainable variable or a trainable variable per step/layer. However, the changes do not yield any significant improvement, but we believe this remains an interesting direction that is worth further investigation with our current setup.
> >
> > **Minor**
> >
> > **Q8 - “On p.9 when introducing terms COV, MIS, and MAT you could refer to their definitions in Appendix G. Also what value does $\delta$ take?”**
> >
> > A8 - Great point! We use the same $\delta$ as ConfGF and CGCF where the threshold is set to 0.5 for GEOM-QM9 and 1.25 for the larger GEOM-Drugs dataset following previous work.
> >
> > **Q9 - “The experiments in Appendix I are interesting and impressive! Maybe it would have been nice to have more explicitly referred to this section in the main paper?”**
> >
> > A9 - Thank you! We have updated the main text to provide more pointers to this section.
> >
> > **Q10 - “How were the hyperparameters described in Appendix E chosen?”**
> >
> > A10 - We try out a small set of hyper-parameters on a validation set, and more details can be found in the updated Appendix E.2.
> >
> > **Q11 - On various grammar/typos.**
> >
> > A11 - Thank you, thank you, thank you!

---

> > > ### Comment · Reviewer_jLPL · 2021-11-22
> > > **Thanks for the rebuttal**
> > >
> > > I thank the authors for their rebuttal! I found the new experiments interesting and helpful, in particular the ablation study showing the significance of the different parts (it actually seems that using a EGNN backbone slightly improves performance...?).
> > >
> > > The other reviewers brought up interesting and important concerns about some of the terminology/statements and the discussion of the related work; these look like they are still somewhat being addressed.
> > >
> > > In terms of my original review, I feel my questions were well answered. I am currently discussing the paper with the other reviewers and will reach back out if anything new comes up!
> > >
> > > p.s. A very minor comment on Figure S3 would be to add datapoint marks  (I wasn't too sure which time steps had actually been tried) and to extend the x axis; also why is the x axis labeled as "number of blocks" rather than e.g. "number of optimization steps"?

---

> > > > ### Author Response · Authors · 2021-11-23
> > > > **Author Response II for Reviewer jLPL**
> > > >
> > > > Thank you for your communication here and the continued engagement throughout the review process :) We are glad that you find our answers helpful, and have addressed other reviewers’ concerns around terminology/statements in our latest text and title updates. Please see below for our responses to the two minor comments you brought up:
> > > >
> > > > **Q1 - Regarding the feedbacks on Figure S3.**
> > > >
> > > > A1 - Thank you for the great suggestions, and we have updated the figure accordingly for clearer visualization.
> > > >
> > > > **Q2 - “it actually seems that using an EGNN backbone slightly improves performance?”**
> > > >
> > > > A2 - The two backbone models are probably on par considering the mixed results and significance of the delta, but it’s indeed interesting to find that the formulation is more important for the generative experiments compared to the parameterization of the SE(3)-equivariant model.

---

> > > > ### Author Response · Authors · 2021-11-29
> > > > **Thank you and about the final score**
> > > >
> > > > Thank you for your reviewing service and continued engagement here. Your reviews have helped us make a stronger and better manuscript, so we really appreciate it. As we are concluding the discussion period and finalizing the review today (Nov. 29th), we want to check in again and see if there are additional concerns we can address for you to consider raising the score as previously suggested. Thanks!

---

> > ### Comment · Reviewer_jLPL · 2021-11-29
> > **End of discussion period update -- increased score.**
> >
> > I thought the authors did a good job addressing my concerns in the rebuttal. I found the new ablation study interesting, and it gave me confidence that the improved performance of the proposed approach over previous work is *not* due to less interesting, minor architecture changes. Furthermore, I thought the new discussion/experiment on the number of time steps required fills in an important detail missing from the original paper. The other reviewers brought up some important concerns about parts of the terminology and related work, and I was glad to see the authors take these on board.
> >
> > Therefore, I have increased my overall score.

---

### Official Review · Reviewer_h2Mm · 2021-11-02

**Correctness:** 3
**Technical Novelty And Significance:** 3
**Empirical Novelty And Significance:** 3
**Recommendation:** 6
**Confidence:** 3

**Main Review:**

Pros:
1. The proposed method seems to be simple and easy to implement.
2. The proposed method outperforms the existing methods in the experiments.

Cons:
1. I think the paper could use some clarifications in their writing. I point out some suggestion in what follows.
- First, I am not sure if the proposed method can be called an "neural energy minimization" method and an "unrolled optimization algorithm." There is no energy or optimization objective for the given DNN architecture. Without any regularization, the proposed DNN architecture is not a proper gradient field.
- The "energy function E()" is defined only on the conformation X in Equation (1), but defined on the conformation X and atom features V in Equation (2).
- This paper uses the terminology "V" as the set of atom nodes and set of atom features. I think the authors should add a new terminology to denote the set of atom features.
- I am a little concerned that the paper propose to solve "conformation optimization", but the proposed architecture is trained using supervised learning and the evaluation is done using RMSD. In other words, there is no optimization in the proposed task. I understand that the task is to "predict the optimized conformation." Maybe the authors could make minor clarifications or changes to clarify this aspect?
2. Additional experimental results can strengthen this paper, especially for future works to build on this paper. For example, can the authors provide their experimental results on predicting the ensemble property (Table 5 in ConfGF paper)?

Minor:
1. Given the training scheme used in the paper, it seems that the DNN-based "molecular conformation generation" methods are applicable to "molecular conformation optimization". Is this correct?
2. How did the authors perform hyper-parameter tuning?

**Summary Of The Paper:**

This paper proposes a new deep neural network (DNN) architecture to generate molecular 3D conformation. It derives its architecture starting from gradient-based updates for minimization of the molecule's energy function. Then the architecture becomes a repetition of vertex-wise aggregation layer of SE(3)-equivariant Transformer-based DNN. The architecture takes a molecule and 3D conformation pre-calculated from RDkit as input. The authors show how the proposed architecture generalize existing DNN architectures for molecular conformation generation. Empirical results demonstrate superiority of the proposed method.

**Summary Of The Review:**

This paper shows solid empirical performance with the proposed DNN architecture for molecular conformation generation. However, there are some logic or details that was not very convincing to me, e.g., the proposed method being a neural energy minimization. I would like to see them clarified for me to raise my score.

---

> ### Author Response · Authors · 2021-11-16
> **Author Response to Reviewer h2Mm**
>
> We thank the reviewer for the feedback and suggestions. Please see below for our responses to the comments.
>
> **Q1 - “I am not sure if the proposed method can be called a ‘neural energy minimization’ method and ‘an unrolled optimization algorithm’. There is no energy or optimization objective for the given DNN architecture. Without any regularization, the proposed DNN architecture is not a proper gradient field.”**
>
> A1 - Regarding “neural energy minimization”, you are correct that we are not explicitly modeling the energy function with the model. Instead, we are estimating the gradient of some energy function with respect to the 3D coordinates with our equivariant neural network. Therefore, the neural network is indeed modeling a gradient field (for coordinates). Since we are training the neural network to slowly move the coordinate of a high-energy conformation (initialization) to a lower-energy stable conformer (reference), the neural network can learn to estimate the gradient from the data and perform the minimization of the conformation energy. In other words, while the model learns to explicitly optimize the conformer, it’s implicitly minimizing the energy by estimating the gradients. We hope the answer here clears the air since this is one of the fundamental concepts of this work, and we are more than happy to improve the presentation if there is any suggestion!
>
> **Q2 - “I am a little concerned that the paper proposes to solve conformation optimization, but the proposed architecture is trained using supervised learning and the evaluation is done using RMSD. In other words, there is no optimization in the proposed task. I understand that the task is to predict the optimized conformation. Maybe the authors could make minor clarifications or changes to clarify this aspect?”**
>
> A2 - As described in Section 3.1, conformation optimization is defined as the process of optimizing an initial estimate of the conformer (usually with higher energy) towards the reference / lowest-energy conformer. Since the model is updating the coordinates at the intermediate step, one could view the process as an unroll optimization on the conformation.
>
> **Q3 - “The energy function $E()$ is defined only on the conformation X in Equation (1), but defined on the conformation X and atom features V in Equation (2).”**
>
> A3 - Thank you for pointing this out! The energy function should depend on both the coordinates (X) and graph features (G) (including atom type and bond type). We have updated the main text to avoid confusion.
>
> **Q4 - “This paper uses the terminology V as the set of atom nodes and set of atom features. I think the authors should add new terminology to denote the set of atom features.”**
>
> A4 - Again, thank you for pointing this out! We have updated the text based on your suggestion.
>
> **Q5 - “Can the authors provide their experimental results on predicting the ensemble property (Table 5 in ConfGF paper)?”**
>
> A5 - Thank you for the great suggestions. The authors of ConfGF have been very kind and provided us with both the code and the 30 examples they used for the evaluation in Table 5 of their paper. We have included the results in Appendix H.2 where we recalculate the performance for RDKit, ConfGF, and ours with their corresponding confidence intervals. Since these are only 30 data points and the confidence intervals are very large, we are not planning to draw any conclusions from the experiment.
>
> **Minor**
>
> **Q6 - “ Given the training scheme used in the paper, it seems that the DNN-based ‘molecular conformation generation’ methods are applicable to ‘molecular conformation optimization’. Is this correct?”**
>
> A6 - In Section 4.2, we argue that a model that works on molecular conformation “optimization” can be extended to solve the “generation” problem as one can always optimize random initialization to an energy stable state following the energy landscape, resulting in different relatively stable conformers. On the other hand, a model that works on generation might not be able to perform optimization for a specific conformer since many of the methods (e.g. CGCF, GraphDG) are not SE(3)-equivariant and therefore not capable to take a conformer as input
>
> **Q7 - “How did the authors perform hyper-parameter tuning?”**
>
> A7 - We try out a small set of hyper-parameter on a validation set, and more details can be found in the updated Appendix E.2.
>
> We would like to thank the reviewer again for the comments, and hope the above response could address the reviewer’s concern.

---

> > ### Comment · Reviewer_h2Mm · 2021-11-19
> > **Thank you for the response.**
> >
> > Thank you for the detailed feedback. I found them to be very helpful.
> >
> > I do believe that the "content" of this paper meets the bar of ICLR. However, I still think the terminology "neural energy minimization" could be misleading (similar to reviewer nNjh and 83EH). This is especially significant since there exists other works that models the energy function more explicitly. If the authors cannot describe which value is being minimized by their architecture, I would like to suggest softening the terminology in general, e.g., include some words like "energy-inspired architecture".
> >
> > I would be happy to raise my scores if (a) the authors reach a consensus with the reviewers on terminology of "energy minimization" and (b) no further critical issues are found during the rest of this rebuttal.

---

> > > ### Author Response · Authors · 2021-11-19
> > > **Thank you for the feedback**
> > >
> > > Thank you for the positive feedback on the improved paper, and the transparency on the scoring. We really appreciate it :)
> > >
> > > Since our formulation is implicitly minimizing the conformational energy by refining the conformer towards its ground state via a neural network, we decided to go with “neural energy minimization” initially in this work. However, we do recognize the confusion this might cause since there is no explicit energy function involved (as found by other readers as well). Therefore, we have updated the terminology around this in our title and main text.
> > >
> > > "Energy-inspired architecture" is a great suggestion, thank you!

---

> > > ### Author Response · Authors · 2021-11-29
> > > **Thank you for raising the score**
> > >
> > > Thank you for raising the score and agreeing the paper meets the bar of ICLR! We are glad that a consensus is reached for the terminology and your suggestions are very helpful.

---

### Official Review · Reviewer_83EH · 2021-11-02

**Correctness:** 2
**Technical Novelty And Significance:** 2
**Empirical Novelty And Significance:** 3
**Recommendation:** 6
**Confidence:** 4

**Main Review:**

Regarding the machine learning approach and experiments, the comparison to existing experiments is unfair at points, since conformer ensembles generated with RDKit are often used as starting guesses, while other approaches rely on random positions. Since the authors have provided experiments with random starting positions in the SI (and still find good, although less impressive performance) this is only a minor issue. However, this difference between model evaluation should be clearly stated in the main text. A discussions on how hydrogen atoms are treated would also be helpful for putting the results into perspective. The presented approach appears to only generate heavy atom positions, with the hydrogens added via RDKit. How does this compare to other approaches and how does it influence the search for stable conformers (e.g. are additional optimizations with ab initio methods necessary)? Since a general correspondence between properties and structure can be expected, the downstream application experiments appear to offer no new insights on the quality of the method. Nevertheless, it would still be helpful to know how the labels for these experiments were generated. Were new computations performed for the new confomers or did the experiments reuse the orginal QM9 labels for molecules with the same graphs? There is also a slight confusion regarding the comparison to CGCF in Table 3. The numbers reported there diverge from [Xu 2020]. In addition, it was never specified which delta value was used for evaluating the COV and MIS scores.

One of the main points of criticism is the proposed neural energy minimization framework for constructing equivariant networks. It essentially recapitulates already established basics of SE(3) equivariant models and offers no new insights. It is well known, that the Cartesian gradients of functions are one way to systematically construct equivariant filters for interactions/updates. This ansatz was e.g. used in  [Schütt 2020] to derive continuous convolutions of an invariant input with an equivariant filter (order l=1). The resulting update has a strong resemblance with Eq 3. In general, updates and interaction can be expressed as a multipole expansion of the energy, and if this is done e.g. in Cartesian coordinates the interaction tensors take the form of first and higher order derivatives w.r.t. atomic positions. Irreducible representations and spherical harmonics are another way to systematically construct equivariant models frequently and are closely related. For example, the update formula in Eq. 3 could also be derived as the interaction between scalar (l=0) and vector (l=1) geometric tensors in this   framework [Batzner 2021]. The statement, that conventional SE(3) equivariant networks are either based on complex mathematical frameworks, which necessitate expensive coefficient calculations, or derived ad hoc from message passing is highly misleading. In general, there is a limited way on how equivariance can be introduced into a model properly. If these operations are to be formulated in a principled and general way, tensor algebra is helpful. This does not automatically imply, that operations are inefficient. If only lower order features/interactions (e.g. l <= 2) are used, no significant computational overhead compared to conventional message passing is introduced (see e.g. [Batzner 2021]). On the other hand, if one were to expand the proposed derivative mechanism to higher order interactions, similar problems would be encountered. E.g. interactions between two dipole vectors require second order derivative tensors (3x3), between quadrupoles a fourth order derivative (3x3x3x3) tensor, etc, thus greatly increasing computational cost.

Finally, there are several problems regarding the introduction and related work sections.
With respect to molecular conformer generation, full ab initio molecular dynamics is neither the standard approach nor required for conformer search. Most of the time, heuristic approaches as e.g. used in RDKit followed by ab initio optimization are used. These are much more tractable and usually work well for small organic molecules, as can be seen based on the performance of RDKit in the conformer generation experiment. The main problem of such approaches is the energetic weighting of the conformers, for which said ab initio optimizations are required.
In the same paragraph (and introduction), a series of generative approaches is described as using distance matrices as intermediates which are then translated into 3D coordinates. However, many of the mentioned approaches do not rely on distance matrices as intermediates. G-SchNet, for example, uses an iterative approach based on factorized probabilities to directly generates a new 3D structure.
The description of ML based force fields is also misleading. While it is true, that sGDML requires retraining for every new molecular graph, this neglects a plethora of machine learning potentials, which do not suffer from this problem and scale linearly with molecule size (see e.g. [Unke 2021] for an overview).
It would also be helpful to provide an in depth discussion on the parallels between ConfGF [Shi 2021] and the present approach in the related works section.


Schütt, Kristof T., Oliver T. Unke, and Michael Gastegger. "Equivariant message passing for the prediction of tensorial properties and molecular spectra." arXiv preprint arXiv:2102.03150 (2021).

Batzner, Simon, et al. "Se (3)-equivariant graph neural networks for data-efficient and accurate interatomic potentials." arXiv preprint arXiv:2101.03164 (2021).

Unke, Oliver T., et al. "Machine learning force fields." Chemical Reviews (2021).

**Summary Of The Paper:**

The authors present an equivariant graph network approach for optimizing and generating molecular conformers.

**Summary Of The Review:**

While sharing conceptual similarities to the gradient fields by [Shi 2021], unrolling the structure optimization process in the form of a graph network is an interesting approach and casting the model as a fixed point search has parallels to deep equilibrium networks. Its potential is demonstrated by good performance in different experiments. The proposed theoretical framework, on the other hand, is a restatement of common concepts in equivariant architectures, which have been presented in a similar or more general form elsewhere. In addition, the different sections (primarily related work) suffer from several fundamental problems which further detract from the quality of the work. Due to the two latter points, I tend towards rejecting the work in its present form.

---

> ### Author Response · Authors · 2021-11-16
> **Author Response to Reviewer 83EH**
>
> We thank the reviewer for the feedback and suggestions, especially your comments around the related work helps us improve the section quite a bit. Please see below for our responses to the comments.
>
> **Q1 - “The comparison to existing experiments is unfair at points since conformer ensembles generated with RDKit are often used as starting guesses, while other approaches rely on random positions.”**
>
> A1 - We don’t think this is true. First of all, in the optimization setting, all models take the same RDKit initialized conformers as the starting point, so there is no unfair advantage there. Second of all, in the generation setting, we have already provided the performances for both random initialization and RDKit initialization. Last but not least, since RDKit initialization is easy to obtain, the fact that our method is able to leverage such informative prior should be considered an advantage. For non SE(3)-equivariant methods like CGCF or GraphDG, they are only taking a graph as input. For ConfGF, they use random initialization because they use a denoising training scheme.
>
> **Q2 - “A discussion on how hydrogen atoms are treated would also be helpful for putting the results into perspective. The presented approach appears to only generate heavy atom positions, with the hydrogens added via RDKit. How does this compare to other approaches and how does it influence the search for stable conformers (e.g. are additional optimizations with ab initio methods necessary)?”**
>
> A2 - Thank you so much for bringing this up! We originally only modeled the heavy atoms in a molecule because prior arts are evaluated only on the heavy atoms. Upon further inspection, we realized the hydrogens are still modeled in methods like ConfGF. Therefore, we have added hydrogen as part of the training in the generative setting and found better performance after considering the spatial constraints of hydrogen. We will update Table 3 with the following values once all the experiments are done.
>
> |  GEOM-QM9               | COV Mean | COV Median | MIS Mean | MIS Median | MAT Mean | MAT Median |
> |-----------------|----------|------------|----------|------------|----------|------------|
> | Ours-Random (w/o Hs) | 88.83 | 93.18 | 30.21 | 30.74 | 0.3778 | 0.3736 |
> | Ours-Random (w/ Hs)     | 91.75 | 95.93 | 30.85 | 33.54 | 0.3462 | 0.3535 |
>
> |  GEOM-Drugs               | COV Mean | COV Median | MIS Mean | MIS Median | MAT Mean | MAT Median |
> |-----------------|----------|------------|----------|------------|----------|------------|
> | Ours-Random (w/o Hs) | 76.50  | 83.78 | 31.40 | 23.03 | 1.0694 | 1.0583 |
> | Ours-Random (w/ Hs)     | 79.90 | 86.82 | 23.37 | 10.39 | 1.0220 | 1.0160 |
>
> **Q3 - “Since a general correspondence between properties and structure can be expected, the downstream application experiments appear to offer no new insights on the quality of the method. Nevertheless, it would still be helpful to know how the labels for these experiments were generated. Were new computations performed for the new conformers or did the experiments reuse the original QM9 labels for molecules with the same graphs?”**
>
> A3 - Regarding the importance of downstream applications, it’s true that there is a correlation between the quality of the structure and the accuracy of the property prediction. It’s not obvious to us at least, how much improvement in structure prediction can be translated into the structure prediction, and hence the experiments. Regarding the label generation, the label for the Figure 3 experiment is generated via Psi4 which is extensively described in the appendix, and the label for Table 2 comes from the QM9 benchmark (Ramakrishnan et al., 2014) as described in the text.
>
> **Q4 - “There is also a slight confusion regarding the comparison to CGCF in Table 3. The numbers reported there diverge from [Xu 2020]. In addition, it was never specified which delta value was used for evaluating the COV and MIS scores.”**
>
> A4 - The results for all baselines are originally reported in the ConfGF paper, where the authors use a different split from CGCF and hence the divergence. We use the same delta following prior art such as CGCF and ConfGF and have updated the values in the appendix. Thank you for pointing this out to us!
>
> **Q5 - “One of the main points of criticism is the proposed neural energy minimization framework for constructing equivariant networks. It essentially recapitulates already established basics of SE(3) equivariant models and offers no new insights.”**
>
> A5 - Regarding your concerns around the novelty of our SE(3)-equivariant parametrization, we want to highlight the main contribution here is not about new models but about providing a new connection between the formulation neural energy minimization and different variants of the SE(3) equivariant models. The keywords here are reinterpreted and extended. To our best knowledge, none of the work you listed explicitly provides such connections.

---

> > ### Author Response · Authors · 2021-11-16
> > **Author Response to Reviewer 83EH - Cont.**
> >
> > **Q6 - “The statement, that conventional SE(3) equivariant networks are either based on complex mathematical frameworks, which necessitate expensive coefficient calculations, or derived ad hoc from message passing is highly misleading.”**
> >
> > A6 - After reading through your comments, we realized some of the wording could mislead readers that are unfamiliar with the field, and have updated a major chunk of the related work section to make it clearer. Thank you for your comments here - we like the current version better as well :)
> >
> > **Q7 - For other comments mostly in the introduction and related sections.**
> >
> > A7 - We have updated the main text to clarify many of the comments there. Thank you for noting them!

---

> > > ### Comment · Reviewer_83EH · 2021-11-19
> > > **Concerns remain**
> > >
> > > I thank the authors for their comments, some concerns have been addressed, some remain.
> > >
> > > First, the revisions to the related work section do not address many of the issues raised above.
> > > Neither ab initio nor conventional molecular dynamics are the standard approach for conformer searches in many applications. Perhaps the authors mean molecular mechanics, since most of the efficient heuristic approaches use empirical force fields?
> > > In addition, machine learning force fields apart from sGMDL are still completely neglected.
> > > As is the fact, that almost all recent machine learning force fields can deal with systems of different size without the need to retrain for every molecules (see suggested review, examples include high dimensional neural network potentials, Gaussian approximation potentials, FCHL Kernel approaches, ANI, SchNet, SOAP, etc)
> > >
> > > In addition, I am still not convinced about the novelty of the theoretical framework or what additional insights it can offer compared to existing approaches.
> > > The use of directional derivatives for expressing equivariance is not novel and has been used in many applications (e.g. multipole expansions for electrostatic interactions in physics). Other prominent examples in machine learning are steerable filters, see e.g. [Freeman 1991] and references within, in particular [Danielsson 1990]. Steerable filters also form the basis of more recent equivariant architectures, see e.g. [Weiler 2018]. The fact, that taking the derivative of a scalar function of the distance yields an equivariant vector and a scalar invariant component and that the latter can be replaced by a non-linear model has also been exploited in other SE(3) equivariant models (e.g. PaiNN, ConfGF). Finally, no systematic manner of constructing filters between different orders of features (with an angular momentum other than l<=1 as is the case for the 3D vectors) are presented, which would be a minimum requirement for a suitable theoretical framework, see e.g. group theory based approaches.  On a side note, without additional adaptations as e.g. in deep equilibrium networks [Bai 2019], the update step is a simple residual update and not a minimization procedure.
> > >
> > > Updating the related work section and focusing on the introduction of the architecture rather than trying to present it as a new theoretical framework would greatly improve the scientific quality of the paper for me. Currently, this detracts from the solid approach and good experimental results.
> > >
> > > Freeman, W. T., & Adelson, E. H. (1991). The design and use of steerable filters. IEEE Transactions on Pattern analysis and machine intelligence, 13(9), 891-906.
> > >
> > > Danielsson, P. E., & Seger, O. (1990). Rotation invariance in gradient and higher order derivative detectors. Computer Vision, Graphics, and Image Processing, 49(2), 198-221.
> > >
> > > Weiler, M., Geiger, M., Welling, M., Boomsma, W., & Cohen, T. (2018). 3d steerable cnns: Learning rotationally equivariant features in volumetric data. arXiv preprint arXiv:1807.02547.
> > >
> > > Bai, S., Kolter, J. Z., & Koltun, V. (2019). Deep equilibrium models. arXiv preprint arXiv:1909.01377.

---

> > > > ### Author Response · Authors · 2021-11-19
> > > > **Author response to remaining concerns of reviewer 83EH**
> > > >
> > > > Thank you for further clarifying your concerns, and recognizing our “solid approach and good experimental results”. We are committed to refining the related work section, so thank you for the continued engagement :) Please see below for our responses to the individual comments.
> > > >
> > > > **Q1 - “Neither ab initio nor conventional molecular dynamics are the standard approach for conformer searches in many applications. Perhaps the authors mean molecular mechanics since most of the efficient heuristic approaches use empirical force fields?”**
> > > >
> > > > A1 - The discussion on the “conventional computational approach” is meant to highlight the fact that there is an accuracy and efficiency tradeoff where you will often need time-consuming ab initio methods for generating very accurate molecular conformers. We understand how this sentence can be confusing to readers on its own, and we will update the text to better reflect our thinkings.
> > > >
> > > > **Q2 - “Machine learning force fields apart from sGDML are still completely neglected. As is the fact, that almost all recent machine learning force fields can deal with systems of different size without the need to retrain for every molecule.”**
> > > >
> > > > A2 - We will make sure to include all the references you pointed out (thank you!), and we believe these slew works are more similar to ConfGF than ours since they are all estimating the “updates” for a single step, while our approach aims to estimate the entire optimization process that is also proven to be more effective in these sample benchmarks based on our empirical results.
> > > >
> > > > **Q3 - Regarding the comments on the novelty.**
> > > >
> > > > A3 - We want to highlight the first two main contributions listed at the end of our introduction section. 1) Our formulation for the conformer prediction problem as an unrolled optimization is novel since prior arts have been mostly focusing on the optimization in a single step (e.g. ConfGF and various NN-estimated force field methods). 2) Our framework explicitly connects an underlying energy function with an SE(3)-equivariant neural network, such that one can potentially derive new models that align with a physical assumption, or interpret some of the existing models from the energy perspective. We never argue this is a new “theoretical” framework or a more general framework that can derive higher-order representations, which might not be practical for our purpose. The complexity of group-theory-based approaches, while could be useful elsewhere, is precisely what we are trying to avoid. With this in mind, we would encourage the reviewer to re-evaluate the novelty claimed by the paper. However, many of the references you pointed out are indeed relevant and we will try to find ways to include them.
> > > >
> > > > **Q4 - “Without additional adaptations as e.g. in deep equilibrium networks [Bai 2019], the update step is a simple residual update and not a minimization procedure.”**
> > > >
> > > > A4 - This is a great point! Ideally, we do want to perform an “infinite” number of optimization steps (i.e. estimating the equilibrium), and this would make a great future direction for this work, so thank you for suggesting it. However, in our current work, we do regularize this to some degree where we feed ground truth conformers as initial estimates with a small probability during the training, such that the model at least knows when equilibrium is achieved. More details on this can be found in E2. In addition, empirical results in Appendix K shows that a large number of steps might not be required for the practical purpose.
> > > >
> > > > **P.S.** We will update the main text after collecting other reviewers’ feedbacks and before the rebuttal period ends.

---

> > > > > ### Comment · Reviewer_83EH · 2021-11-22
> > > > > **Response**
> > > > >
> > > > > Thank you for the additional comments. I believe incorporating aspects of the answer to Q3 in the main text would help to clarify what is meant with framework in this context. If this and the issues regarding the related work section and terminology are addressed, I am willing to raise my score.

---

> > > > > > ### Author Response · Authors · 2021-11-23
> > > > > > **Author Response III for reviewer 83EH**
> > > > > >
> > > > > > Thank you for your continued engagement throughout the process, and we really appreciate the positive attitude:)
> > > > > >
> > > > > > To address your concerns, we have updated the text to:
> > > > > > 1) better clarify our contribution in the intro and related work section in relation to Q3
> > > > > > 2) improve the statements around the “conventional computation approach"
> > > > > > 3) include other machine learning force fields methods in the related work section for better context
> > > > > > 4) update the terminology (including titles) around “energy minimization” for less confusion.
> > > > > >
> > > > > > We hope these address your concerns and are committed to making additional changes in the text post the rebuttal period if needed.

---

> > > > > > ### Author Response · Authors · 2021-11-29
> > > > > > **Thank you and about the final score**
> > > > > >
> > > > > > Thank you for your reviewing service and continued engagement here. Your reviews have helped us make a stronger and better manuscript, so we really appreciate it. As we are concluding the discussion period and finalizing the review today (Nov. 29th), we want to check in again and see if there are additional concerns we can address for you to consider raising the score as previously suggested. Thanks!

---

> ### Author Response · Authors · 2021-12-01
> **Further comments from reviewer 83EH?**
>
> We have found the comments here very helpful, and want to check in again and see if there are further comments that we can address for you to raise the score as previously suggested in the thread. Thanks!

---

> > ### Author Response · Authors · 2021-12-01
> > **Thank you for raising the score**
> >
> > Thank you for raising the score, and we appreciate your engagement throughout the rebuttal period. They are very helpful!

---

### Official Review · Reviewer_nNjh · 2021-11-03

**Correctness:** 3
**Technical Novelty And Significance:** 3
**Empirical Novelty And Significance:** Not applicable
**Recommendation:** 3
**Confidence:** 4

**Main Review:**

Concerns about technical parts
- My major concern is the formulation of "energy minimization". Although the main claim of the work is energy minimization, I don't find any relation between the mathematical formulation and energy minimization.
    - The potential energy function $u()$ in eqn (2) is not specified.
    - $f_x()$ in eqn (3) is not trained to minimize the energy function (which is not specified); instead, it is trained to minimize the L2 distance between the generated conformer and its closest reference conformer.
- The basic idea of this work needs justifications. The authors seem to suggest that casting "the prediction problem into an unrolled optimization process" is better than previous approaches such as ConfGF. This is not clear to me.
   - "While our model is also trying to estimate the gradient field, we instead employ an unrolled optimization architecture (Domke, 2012; Liu et al., 2018) aiming to model the entire optimization process. By learning to minimize a parametric energy function in a fixed number of steps, our models enjoy a much faster inference time."  This claims that the proposed method is faster but not better (or more accurate). Then it is better to explain why it is better than ConfGF in Table 3, actually much much better in terms of MIS(%).
    - Seems the number of steps is a key hyperparameter in the proposed method, which is unfortunately not studied in experiments. For example, how many steps are used in experiments? Are the final results sensitive to the number of steps?

Concerns about experiments
- In Section 4.1, Table 1 shows that the three atom model is better than the two atom model for conformation optimization. However, only the two atom model is studied and compared in Section 4.2 for conformation generation. I expect the comparison with the three atom model rather than the two atom model in Section 4.2.
- As shown in Table S1, the results of the proposed method are (very) sensitive to the initialization of molecular conformation, i.e., Ours-TwoAtom is much better than Ours-TwoAtom-Random. Does this mean the proposed model can only work well with a good initialization? If yes, this may limit the application scope of the proposed model.
- Similar concern about "instead of training the model to optimize a single initialization towards one reference, we train the model to optimize K initialization such that these K optimized conformers can be exactly matched to K different sampled references." Are the K initializations different from each other? If so, how to get the K different initializations?

**Summary Of The Paper:**

This work studies the problem of molecular conformation optimization and proposes a neural "energy minimization" formulation, where a neural network is parametrized to learn the gradient fields of a conformational energy landscape.

**Summary Of The Review:**

I have several concerns about the technical part and empirical results.

---

> ### Author Response · Authors · 2021-11-16
> **Author Response to Reviewer nNjh**
>
> We thank the reviewer for the feedback and suggestions. Please see below for our responses to the comments.
>
> **Q1 - “Although the main claim of the work is energy minimization, I don't find any relation between the mathematical formulation and energy minimization.”**
>
> A1 - In the first part of the method (Section 3.2), we have pointed out the relationship between energy minimization and molecular conformation optimization. Since the energy here refers to the conformation energy and different conformations can result in different conformation energy, the process of optimizing the molecular coordinates towards more energy preferred state via gradient descent can be viewed as energy minimization. We hope the answer here clears the air since this is one of the fundamental concepts of this work, and we are more than happy to improve the presentation if there is any suggestion!
>
> **Q2 - “The potential energy function $\mu$ in eqn(2) is not specified.”**
>
> A2 - Yes, and that’s precisely the point of “neural” energy minimization. In normal energy minimization via gradient descent, one would fully specify an energy function and perform gradient descent. However, such specifications might be 1) non-differentiable with respect to the coordinates, and 2) the heuristically designed energy function might not be accurate. Therefore, we propose to use a neural network to directly model the coordinate gradient itself (thanks to the equivariant setup), and learn to minimize the energy (or optimize the conformer) from the data. In eqn(2), we only specify the parametrization of such energy function, and the neural network is tasked to learn the exact derivative of such energy function.
>
> **Q3 - “$f_x$ in eqn (3) is not trained to minimize the energy function; instead, it is trained to minimize the L2 distance between the generated conformer and its closest reference conformer.”**
>
> A3 - As pointed in A1, since we are training the model to optimize “a” conformation toward the most energy stable reference conformer, it’s equivalent to training the model to minimize the conformation energy of a conformer.
>
> **Q4 - “... claims that the proposed method is faster but not better (or more accurate). Then it is better to explain why it is better than ConfGF in Table 3, actually much much better in terms of MIS(%).”**
>
> A4 - The intention of this statement is not to say “faster inference” is the only benefit of the proposed method against ConfGF. Rather, it serves as a motivation for why we choose to model the entire optimization process as opposed to a single-step optimization which would rely on computationally-intensive sampling methods like Langevin dynamics. We have updated the text in related work to better reflect our thinking and conducted ablation studies to show why our model is better as below, which is also attached in Appendix J.
>
> |               | COV Mean | COV Median | MIS Mean | MIS Median | MAT Mean | MAT Median |
> |-----------------|----------|------------|----------|------------|----------|------------|
> | Multi-step Eval | 72.96    | 76.73      | 37.78    | 28.88      | 1.1031   | 1.0853     |
> | EGNN-bb         | 77.09    | 82.57      | 23.88    | 11.58      | 1.0491   | 1.0514     |
> | Non OT-Loss     | 59.67    | 60.96      | 16.41    | 1.82       | 1.2443   | 1.2253     |
> | Full model      | 76.50    | 83.78      | 31.40    | 23.03      | 1.0694   | 1.0583     |
>
> **Table A**: Conformation Generation Ablation Studies on the GEOM-Drugs Dataset.
>
> We find the formulation of framing the generative task as a fixed-step unroll optimization process using an SE(3)-equivariance model is the more important bit, as we only see a very small performance regression when switching our backbone optimization model to EGNN. On the other hand, ConfGF is proposed to model one-step optimization and relies on Langevin dynamics for performing the entire sequence of optimization. Additionally, the optimal transport loss also helps quite a bit, as it encourages the model to optimize the conformers towards different directions instead of collapsing the mode.
>
> **Q5 - “Seems the number of steps is a key hyperparameter in the proposed method, which is unfortunately not studied in experiments. For example, how many steps are used in experiments? Are the final results sensitive to the number of steps?”**
>
> A5 - This is a great suggestion! In the updated Appendix K, we added a new experiment where we test the optimization performance on the QM9 dataset with different numbers of optimization steps. We observe a diminishing return for steps > 4 since the “gradient” would be small if we are close to the solution. Therefore, we believe the method is not sensitive to the number of steps as long as a sufficient number of steps are taken. In all our experiments, we perform a fixed number of 9 optimization steps. Again, thank you for this great suggestion, and we have also updated the main text to highlight the finding.

---

> > ### Author Response · Authors · 2021-11-16
> > **Author Response to Reviewer nNjh - Cont.**
> >
> > **Q6 - “I expect the comparison with the three atom model rather than the two atom model in Section 4.2.”**
> >
> > A6 - As pointed out at the beginning of Section 4, the purpose of the experiment/benchmark in the generative setting (Section 4.2) is to show that our proposed optimization formulation can be easily extended from the optimization setting to the generative setting, and achieve comparable performances with the state-of-the-art. Since the claim here is more about the versatility of the optimization formulation, we just use the simpler two-atom model here and didn’t even change the hyper-parameter following the same values from the optimization experiment. However, one could certainly expect some level of performance gain by switching to the three-atom model and performing proper hyper-parameter tuning, but that’s not what we are after here.
> >
> > **Q7 - “Does this mean the proposed model can only work well with a good initialization? If yes, this may limit the application scope of the proposed model.”**
> >
> > A7 - This is not true. First of all, the experiment in Table S1 is performed exactly to counter this point. Even with the random initialization, the optimization framework is still able to predict conformer much better than the RDKit initialization in the optimization setting. However, being able to build upon prior knowledge and a good initialization can certainly further improve the performance, not to mention these initial estimates are easy to obtain. Therefore, this should be viewed as advantages instead of disadvantages, as other non-se3-equivariant methods (e.g. CGCF) are not able to utilize such prior naturally. Last but not least, in the generative setting, we find random initialization sometimes can work better than the RDKit initialization for larger molecules (potentially) due to the limitation of RDKit’s sampling procedure.
> >
> > **Q8 - “Are the K initializations different from each other? If so, how to get the K different initializations?”**
> >
> > A8 -  Yes, the K initializations are different and that’s why we can generate a diverse set of conformers through the optimization process. In the generative experiment Table 3, Ours-Random is initialized with K set of random coordinates sample from a Gaussian distribution, while Ours-RDKit is initialized with K set of coordinates sample from RDKit with a different random seed. More details can be found in Appendix G.

---

> > > ### Author Response · Authors · 2021-11-23
> > > **Other concerns?**
> > >
> > > We want to check in again and see if there are additional concerns we haven’t addressed yet for a better score. Noted we are not able to make main text updates after Nov. 22nd. However, we are happy to continue the discussion here and incorporate additional feedbacks post the rebuttal period :)

---

> > > ### Comment · Reviewer_nNjh · 2021-11-26
> > > **Regarding initialization**
> > >
> > > > Even with the random initialization, the optimization framework is still able to predict conformer much better than the RDKit initialization in the optimization setting.
> > >
> > > If I'm right, "the RDKit initialization in the optimization setting" means RDKit+MMFF in the table. I don't think we can draw the conclusion that
> > > >  the experiment in Table S1 is performed exactly to counter this point.
> > >
> > > As the proposed method is ML-based and leverages labeled data for model training, it should not be compared with no ML-based methods such as RDKit+MMFF. At least, some previous deep learning-based methods (e.g., ConfGF) should be compared to justify that the proposed method works well with random initiallizaiton.

---

> > > > ### Author Response · Authors · 2021-11-27
> > > > **Author response**
> > > >
> > > > Thank you for the comment! Before we go into the details of the specific questions, we want to focus on the following points regarding your concerns around the initialization:
> > > >
> > > > * Our method works well with random initialization.
> > > >   * In the optimization setting (Table S1), it is still able to arrive at a reasonable (i.e. better than using RDKit + MMFF) estimate from random coordinates.
> > > >   * In the generative setting (Table 3), **our method with random initialization** is also able to outperform other random initialized methods (e.g. ConfGF) in many of the datasets/metrics. In some of the cases, random initialization can even outperform the RDKit initialization due to better sampling of the space.
> > > >   * Therefore, our method works well with random initialization and we do not agree with your assessment that “the proposed model can only work well with a good initialization”.
> > > > * Since our method works well with random initializations, the fact that our method is capable of leveraging prior estimates to get better performance should not be treated as a limitation but as an advantage over methods like CGCF.
> > > >
> > > > **Q1 - “At least, some previous deep learning-based methods (e.g., ConfGF) should be compared to justify that the proposed method works well with random initialization.”**
> > > >
> > > > A1 - TableS1 shows the result for the optimization setting, while ConfGF is designed/trained for the generative setting. However, we did compare the two methods in the generative setting (Table 3) with random initialization and ours work quite well.
> > > >
> > > > **Q2 - “As the proposed method is ML-based and leverages labeled data for model training, it should not be compared with no ML-based methods such as RDKit+MMFF.”**
> > > >
> > > > A2 - The point here is to show that our method can arrive at a reasonable conformer after optimizing from random initialization, and it’s “reasonable” because it’s more accurate than the one from RDKit+MMFF in the benchmark. The conclusion has never been that an ML-based method can replace a general force field method, or is better in an out of domain setting.

---

> > > > > ### Comment · Reviewer_nNjh · 2021-11-28
> > > > > **Table 3**
> > > > >
> > > > > Thanks for pointing out Table 3.
> > > > > > In the generative setting (Table 3), our method with random initialization is also able to outperform other random initialized methods (e.g. ConfGF) in many of the datasets/metrics. In some of the cases, random initialization can even outperform the RDKit initialization due to better sampling of the space.
> > > > >
> > > > > I'm not sure whether we can draw the conclusion that "our method with random initialization is also able to outperform ConfGF in many of the datasets/metrics". Of the two datasets in this table, Ours-Random is better on only one dataset.
> > > > > 1. On GEOM-QM9, Ours-Random performs similarly to ConfGF: in terms of COV metrics, they are similar; for MIS, Ours-Random is much better; for MAT, ConfGF is much better.
> > > > > 2. On GEOM-Drugs, Ours-Random is much better.
> > > > >
> > > > > I'm not very familiar with the two datasets. I have a few further questions:
> > > > > 1. Why is Ours-Random perform significantly better on the Drugs dataset than the QM9?
> > > > > 2. In Table S1, Ours-TwoAtom-Random is universally much worse than Ours-TwoAtom on both datasets (this is why I think the proposed method is sensitive to initialization), while in Table 3, they perform similarly on the Drugs dataset. Could you explain the inconsistency?

---

> > > > > > ### Author Response · Authors · 2021-11-29
> > > > > > **Response to Questions of Table 3**
> > > > > >
> > > > > > **Q1 - “In Table S1, Ours-TwoAtom-Random is universally much worse than Ours-TwoAtom on both datasets (this is why I think the proposed method is sensitive to initialization), while in Table 3, they perform similarly on the Drugs dataset. Could you explain the inconsistency?”**
> > > > > >
> > > > > > A1 - It seems that there is some confusion on our experiment settings. As described at the beginning of the experiment section, we evaluated the methods in two different settings with separated goals: 1) an optimization setting where we aim to recover one or a few most stable conformers as accurate as possible (measured by RMSD), and 2) a generative setting where we aim to capture the distribution of relatively stable conformers measured by “recall” (COV), “precision” (MIS), and “quality” (MAT).
> > > > > >
> > > > > > Table S1 shows the performance for Setting 1 while Table 3 shows the performance for Setting 2, and different settings are trained differently as well, so one should not directly compare the results from the two. However, we do see a pattern in the performance where RDKit initialized conformers can produce more accurate and higher quality conformers compared to random initialized (measured by RMSD in Setting 1 and MIS/MAT in Setting 2). The “inconsistency” here might refer to random initialization having better COV scores for Table 3 while observing lower RMSDs in Table S1, but they are different metrics measuring different properties of the methods and we have discussed the differences/tradeoffs in the Results paragraph on page 9.
> > > > > >
> > > > > > **Q2 - “Why is Ours-Random perform significantly better on the Drugs dataset than the QM9?”**
> > > > > >
> > > > > > A2 - Since GEOM-Drugs and GEOM-QM9 are different datasets, we are not sure what “better” means here.
> > > > > > * If “better” here means comparing to Ours-RDKit: First of all, while it’s true that Ours-Random performs significantly better on the COV score in the GEOM-Drugs dataset, it’s still behind in terms of the median MIS/MAT score as expected. Secondly, as discussed in the text, GEOM-Drugs contain larger molecules with more rotatable bonds compared to the GEOM-QM9 datasets. Therefore, the stable conformers in GEOM-Drugs dataset can be much more diverse and challenging for RDKit initialized conformers to sufficiently cover.
> > > > > > * If “better” here means comparing to ConfGF: It’s probably because ConfGF can be trained to accurately estimate gradients of atom coordinates on the small molecule dataset like GEOM-QM9, and thus it can benefit from a large number of langevin dynamic steps to refine conformations precisely and achieve better MAT score. While for large molecule datasets like GEOM-Drug, the one-step gradient estimation of ConfGF is not accurate enough, so the conformations may not be refined towards reference ones during the inference phase. Compared to ConfGF, **our model learns the whole optimization process instead of the one-step optimization.** Although our model can not refine conformations of small molecules as fine as ConfGF due to less update steps (appear as worse MAT on GEOM-QM9), it can better model the whole optimization process and make most initialized random conformers optimized towards one of the correct directions. Therefore,  our model can achieve better MIS scores and also other scores especially on the more challenging GEOM-Drug dataset.

---

> > > > > > > ### Author Response · Authors · 2021-12-01
> > > > > > > **Other comments?**
> > > > > > >
> > > > > > > Hope we have clarified the misunderstanding here and please let us know if you have further questions.

---

> > ### Comment · Reviewer_nNjh · 2021-11-26
> > **Number of steps**
> >
> > Thanks for the update. Well addressed.

---

### Decision · Program_Chairs · 2022-01-20

**Decision:**

Accept (Poster)

**Comment:**

All reviewers except one agreed that this paper should be accepted because of the strong author response during the rebuttal phase. Specifically the reviewers appreciated the new ablation study showing that improvements are not due to minor architectural changes, the new experiment on the number of time steps required for experiments, the agreement to change language around "neural energy minimization", the improvements to the related work, the novelty of the unrolled optimization approach, and the nice experimental results. Given this, I vote to accept. Authors: please carefully revise the manuscript based on the suggestions by the reviewers: they made many careful suggestions to improve the work and stressed that the paper should only be accepted once these changes are implemented. Once these are done the paper will be a nice addition to the conference!